# MEXMA: TOKEN-LEVEL OBJECTIVES IMPROVE SENTENCE REPRESENTATIONS

## ABSTRACT

Cross-lingual sentence encoders (CLSE) create fixed-size sentence representations with aligned translations. Current pre-trained CLSE approaches use sentence-level objectives only. This can lead to loss of information, especially for tokens, which then degrades the sentence representation. We propose MEXMA, a novel approach that integrates both sentence-level and token-level objectives. The sentence representation in one language is used to predict masked tokens in another language, with both the sentence representation and *all tokens directly updating the encoder*. We show that adding token-level objectives greatly improves the sentence representation quality across several tasks. Our approach outperforms current pre-trained cross-lingual sentence encoders on bitext mining as well as several downstream tasks. We also analyse the information encoded in our tokens, and how the sentence representation is built from them.

## 1 INTRODUCTION

Creating general-purpose multilingual embeddings has attracted significant attention from the research community in recent years, driven by the growing need for efficient and effective cross-lingual representations. Cross-Lingual Sentence Encoders (CLSE) create fixed-size sentence representations that are able to capture the relevant information in a sentence, and are aligned across languages. By capturing relevant sentence information in a shared multilingual space, these aligned representations enable *efficient* comparison and retrieval based on distance measures, thereby facilitating their effective utilization in various downstream applications.

Current CLSE (Duquenne et al., 2023; Feng et al., 2022) typically build upon pre-trained encoders, often language models (Conneau et al., 2020; Devlin et al., 2019) or translation models (NLLB Team et al., 2022). These pre-trained encoders have been trained using objectives that focus on individual words or tokens, i.e. token-level objectives. Examples of such objectives include unmasking, where the model is required to predict each token individually, and *all predictions* are used to *update the encoder* directly. However, Muennighoff et al. (2023); Hu et al. (2020) show that pre-trained encoders without objectives that consider entire sentences, i.e. sentence-level objectives, do not create good sentence representations. This means that CLSE need to train using sentence-level objectives, in order to effectively capture the relevant information of the sentences.

Although CLSE start from encoders pre-trained with token-level objectives, they are commonly trained with sentence-level objectives that *only update the encoder through the sentence representation* (Duquenne et al., 2023; Feng et al., 2022; Yang et al., 2019; Artetxe & Schwenk, 2019a), without any objective for each token individually. We hypothesize that token-level objectives should be kept during the training of CLSE, coupled with the sentence-level objectives, to better update the encoder and improve sentence representation quality and alignment. The intuition is that only using sentence-level objectives leads to a degradation of token level information, especially lexical information, which in turn can impact the sentence representation.

Recently, there have been approaches exploring the use of both token-level and sentence-level objectives for better sentence representations. In DAP (Li et al., 2023), the token-level objective is only used to update the token representations in the encoder, without influencing directly the sentence representation. In RetroMAE (Xiao et al., 2022), the tokens are not directly updated with the same token-level objective as the sentence representation.

To effectively combine token and sentence-level objectives, we propose MEXMA, a new approach that uses the sentence representation in one language to predict masked tokens in another language, and uses both the sentence and tokens' information to update the encoder. This token-level objective is combined with a sentence-level objective to enforce sentence alignment across languages.

Our approach outperforms state-of-the-art pre-trained cross-lingual sentence encoders, namely LaBSE and SONAR, on several key tasks including bitext mining, classification, and pair classification. Specifically, we report notable gains on the xsim++ benchmark computed over the FLO-RES200 test set, where MEXMA achieves an error rate of 9.6%, surpassing SONAR's 12.1%. Additionally, in classification tasks evaluated on MTEB and SentEval, MEXMA achieves an accuracy of 65.4% compared to SONAR's 63.0%. The larger supervision in MEXMA enables training smaller models with better alignment than LaBSE ($\approx 2\times$) and close to SONAR's performance ($\approx 3\times$).

Our main contributions are:

- We introduce a novel architecture leveraging both sentence-level and token-level objectives outperforming current approaches.
- We perform ablation studies that show the impact of token-level objectives on the sentence-level representations performance.
- We provide an extensive analysis of the inner working of our model, by analysing its tokens' contents, and the way the sentence embedding is built. We show that as a byproduct of our training, individual tokens are also well aligned across languages.
- We show that our approach can also be coupled with existing alignment approaches, specifically contrastive learning, and improve its quality.
- Our code and model are available here: HIDDEN FOR ANONYMITY

## 2 RELATED WORK

Sentence embeddings have been well studied in the last decade. Initially, recurrent networks were trained to predict previous and next sentence (Kiros et al., 2015) or sentence entailment (Conneau et al., 2017). Universal Sentence Encoder (Cer et al., 2018) trains a transformer network on both tasks. Reimers & Gurevych (2019) propose to continue the training of a BERT model to include a sentence-level objective. These initial works have been extended to multilingual settings, to capture the relevant information in the sentences, and to have aligned representations across languages. These new approaches are called cross-lingual sentence encoder. We describe those works next.

**UPDATE VIA SENTENCE REPRESENTATION** Most current cross-lingual sentence encoder approaches only update their encoder via the sentence representation objective, without having any token-level objective in the output of the encoder that would update each token individually (Guo et al., 2018; Yang et al., 2019; Feng et al., 2022; Artetxe & Schwenk, 2019a; Duquenne et al., 2023; Heffernan et al., 2022). They are most commonly based on contrastive learning (Hadsell et al., 2006) methods, that aim to reduce the distance between positive pairs (translations) and increase the distance between negative pairs (non-translations) (Guo et al., 2018; Yang et al., 2019; Feng et al., 2022). Notably, LaBSE (Feng et al., 2022) uses the contrastive loss, with the additive margin softmax approach of Yang et al. (2019). Non-contrastive approaches reduce the distance between positive pairs (translations) only, being prone to *collapse*. A common solution to collapse is to use an auto-regressive decoder to prevent it. For CLSE, it is common to use translation (Artetxe & Schwenk, 2019a; Duquenne et al., 2023) with a fixed-size sentence representation after the encoder (bottleneck), assuming that a model can translate a sentence into many languages only if a good sentence-level conceptual representation is learned. The bottleneck, however, prevents gradients from the decoder to directly update the individual token representations of the encoder, which we hypothesize leads to a degradation of token level information and consequently of the sentence representation. Our method also uses a sentence representation as context for the unmasking, but allows direct token-level gradients to propagate to the encoder token representations.

**UPDATE VIA SENTENCE AND TOKEN REPRESENTATIONS** Recent approaches (Li et al., 2023; Xiao et al., 2022; Wei et al., 2021; Fan et al., 2022) have shown that combining token and sentence level objectives can improve sentence representations. RetroMAE (Xiao et al., 2022), is an Infor-

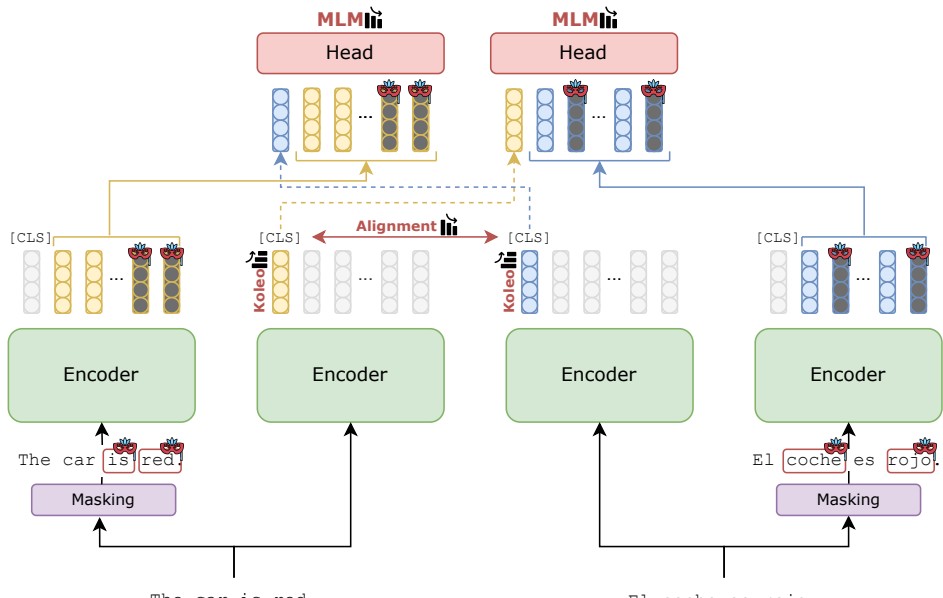

Figure 1: MEXMA architecture. Given two translations, we create two views for each, a masked and a clean version (symmetrical architecture), and use the sentence representations from one language to unmask the other (cross-unmasking). We align the clean sentence representations via the alignment loss, and increase the usage of the space with the KoLeo loss.

mation Retrieval method that utilizes fixed-size sentence representations to guide token unmasking, demonstrating its effectiveness in enhancing sentence representation quality. The encoder itself is only updated by its own MLM loss with light masking, and via the gradients coming from the sentence representation, but not from the direct token-level gradients of the heavy unmasking with the sentence representation as context. The masking in the encoder input forces the sentence representation to not be built from the full sentence, which is not ideal. Wei et al. (2021) combines MLM with the contrastive loss. However, the alignment between sentences is performed on masked sentences, and the unmasking is not done with a cross-lingual sentence context. DAP (Li et al., 2023) proposes to jointly align tokens and sentence representations. It performs unmasking with all tokens of the other language as context, which means it updates the encoder with each token individually. However, it relies exclusively on the contrastive loss to update the sentence representations, and the sentence representation is not used to perform the token unmasking. In our work, we show that sentence and token-level objectives can be much more intertwined, with both individual tokens and the sentence representation updating the encoder, and each other, leading to improved performance.

Detailed diagrams of the described architectures are provided in Appendix H.

## 3 METHODOLOGY

We propose MEXMA, a novel multilingual alignment technique based on both token-level and sentence-level objectives. The goal is to create a sentence representation that is able to encode the syntactic, semantic and lexical information in a sentence, with representations well aligned across languages. To achieve this goal, inspired by monolingual masked auto-encoding techniques (Xiao et al., 2022), we use the sentence representation in one language to unmask the tokens in another language, updating both the sentence and individual tokens. This also forces the sentence representation to encode the relevant parts of the sentence. Using masking also allows us to use a non-contrastive loss to align sentence representations, since it prevents the collapse. Both sentence and token-level objectives are used to improve the quality of the sentence representation. Our architecture is depicted in Figure 1, and is composed of several components that we describe now. For the explanation, we refer to inputs, models and outputs that have no masking as *clean*, and *masked*

for their masked counterparts. Additionally, we consider two languages, language $\mathcal{A}$ and language $\mathcal{B}$, which are associated with the sentence representations $S_{\mathcal{A}}$ and $S_{\mathcal{B}}$ (from the clean encoders).

**THE CROSS-UNMASKING**    To ensure that our sentence vector captures the meaningful information of the sentence, we mask a significant portion of the input tokens in language $\mathcal{A}$. This makes it challenging for the encoder and the MLM head to recover the missing tokens without any additional context. To overcome this challenge, we provide the unmasking head with the sentence vector $S_{\mathcal{B}}$, derived from the clean sentence in language $\mathcal{B}$. This forces the model to leverage the information in $S_{\mathcal{B}}$ to predict the masked tokens in language $\mathcal{A}$. By doing so, we encourage the sentence vector to capture the essential information of the sentence. Furthermore, by alternating languages, we enforce the sentence vector to encode information that is useful across languages. We formulate this component into a symmetrical cross-entropy loss (CE), applied over the outputs of the encoders:

$$\mathcal{L}_{mlm} = CE([S_{\mathcal{B}}, \hat{A}], A) + CE([S_{\mathcal{A}}, \hat{B}], B),$$

where $\hat{A}$ and $\hat{B}$ are the outputs of the masked encoders without the CLS embedding, A and B the masked tokens' targets, and $[X, Y]$ represents the concatenation of X and Y.

**THE ALIGNMENT LOSS**    The cross-unmasking generates an implicit alignment due to the switching of languages to perform the unmasking. However, as is, that implicit alignment does not strongly enforce the same sentence representations in two different languages to be close in the embedding space. Following SONAR (Duquenne et al., 2023), to further reinforce the spatial proximity of semantically equivalent sentences across languages, we use an additional non-contrastive alignment objective. The two losses, unmasking and alignment, complement each other to provide both aligned and meaningful vector representations of sentences in multiple languages. We formulate this component as a Mean Squared Error (MSE) loss between sentence representations:

$$\mathcal{L}_{alignment} = MSE(S_{\mathcal{A}}, S_{\mathcal{B}}),$$

**THE SYMMETRICAL ARCHITECTURE**    To align all languages and maximize data usage, we adopt a symmetrical approach that unmasks the tokens of language $\mathcal{A}$ with $S_{\mathcal{B}}$, and vice versa, simultaneously. We thus create four instances of the encoder (with shared parameters). For each language, we have two versions of each sentence: one heavily masked and one clean. This allows us to generate two clean sentence vectors, $S_{\mathcal{A}}$ and $S_{\mathcal{B}}$, which is essential for aligning representations between languages. A non-symmetrical approach with only two encoders (one per language) would not produce the desired alignment as it would force the model to align a heavily masked sentence vector with a clean one, which is not ideal.

**THE KOLEO LOSS**    In preliminary experiments, we noticed that our representations exhibited more anisotropy than those learned with contrastive approaches. This has been shown to impact the quality of the representations (Godey et al., 2024). Inspired by DINOv2 (Oquab et al., 2024), we employ the KoLeo loss (Sablayrolles et al., 2019) to encourage sentence representations to spread out evenly in the latent space. The KoLeo loss is based on the Kozachenko-Leonenko differential entropy estimator (see Beirlant et al. (1997)). We define below the KoLeo loss, $L_{KoLeo}$, for a set of $n$ representations, as well as the symmetrical version, $L_K$, we use to train our models:

$$\mathcal{L}_K = \mathcal{L}_{KoLeo}(S_{\mathcal{A}}) + \mathcal{L}_{KoLeo}(S_{\mathcal{B}}) \quad \text{with} \quad \mathcal{L}_{KoLeo} = -\frac{1}{n} \sum_{i=1}^{n} log(d_{n,i})$$

where $d_{n,i} = min_{j \neq i} \parallel x_i - x_j \parallel$ is the distance between $x_i$ and its nearest point in the batch.

Our training loss is a weighted combination of all previous losses:

$$\mathcal{L}_{MEXMA} = \alpha \cdot \mathcal{L}_{alignment} + \beta \cdot \mathcal{L}_{mlm} + \gamma \cdot \mathcal{L}_K$$

where $\alpha$, $\beta$ and $\gamma$ are hyper-parameters that control the weight of each loss term. To show that MEXMA can be used on top of existing alignment approaches, we provide, in Section 5.2, experimental results when replacing the MSE alignment loss in MEXMA with a contrastive loss.

| Model | xsim ↓ | xsim++ ↓ | BUCC ↑ | o-xsim ↓ | o-xsim++ ↓ | d-xsim ↓ | d-xsim++ ↓ |
|-------|--------|----------|--------|----------|------------|----------|------------|
| DAP | - | - | 98.68 | - | - | 2.90 | 32.82 |
| SONAR | 0.09 | 12.08 | 98.25 | 0.08 | 11.68 | 0.04 | 10.55 |
| LaBSE | 0.92 | 18.65 | 98.75 | 0.31 | 16.21 | 0.26 | 14.51 |
| MEXMA | **0.06** | **9.60** | **98.93** | **0.05** | **9.01** | **0.02** | **8.26** |

Table 1: Results in mining (%). xsim and xsim++ are computed on 81 languages (FLORES200 dataset, X-eng pairs), with o-. . . columns showing results for 72 supported languages from LaBSE and d-. . . columns showing results for 34 languages supported by DAP. BUCC is computed with F1 in its 4 languages.

### 3.1 EXPERIMENTAL SETUP

**ENCODER BACKBONE**   As our encoder, we utilize a modified version of the XLM-RoBERTa model (Conneau et al., 2020) provided by HuggingFace that uses a more efficient attention (details in Appendix A). Our sentence representation from the encoder is obtained via the CLS embedding of the last layer, without any further processing.

**TRAINING DATA**   Our training dataset is a subset of the NLLB-200 corpus (NLLB Team et al., 2022), which comprises 200 languages. We cover 81 languages, utilizing only publicly available data, all sourced from Opus (Tiedemann, 2012). The specific languages used are listed in Appendix C. We always train using one sentence in English associated with its translation in one of the remaining 80 languages, as done in SONAR. The dataset consists of a combination of human-translated and synthetic data, where we attempt to impose a minimum of 15 million sentences per language. For languages with limited human-annotated data, we supplemented the dataset with mined data from NLLB (Schwenk et al., 2020; Fan et al., 2020; NLLB Team et al., 2022) to reach the 15 million sentence threshold. Conversely, to ensure that our dataset is somewhat balanced across languages, for languages with abundant human-annotated data, we capped the dataset at 25 million sentences per language. The datasets used are detailed in Table 17.

We provide additional details about the parameters and configurations of our model in Appendix A.

## 4 RESULTS

To assess the quality and alignment of our embeddings, we evaluate them on a range of tasks. These tasks fall into two categories: mining tasks and other downstream tasks. Mining tasks measure how aligned our representations are across languages, while downstream tasks evaluate the generalization power and overall quality of our embeddings.

### 4.1 MULTILINGUAL ALIGNMENT THROUGH MINING

We evaluate our model on three alignment tasks: xsim[1], xsim++ (Chen et al., 2023), and BUCC (Zweigenbaum et al., 2018; 2017). Both xsim and BUCC involve retrieving the correct translation of a query sentence from multilingual datasets. xsim++ adds complexity by introducing hard negatives in English sentences. Following Heffernan et al. (2022), we exclude Tatoeba due to limited data and low-quality translations.

xsim and xsim++ use a margin-based similarity approach (Artetxe & Schwenk, 2019b), while BUCC employs cosine similarity. xsim and xsim++ scores are the error rate of misaligned sentences, whereas BUCC uses the F1 score, evaluated with the MTEB benchmark (Muennighoff et al., 2023).

BUCC covers German, French, Russian and Chinese. We evaluate our model using xsim and xsim++ on the FLORES200 dataset, covering the 81 languages supported by our model (listed in Appendix C). For fairer comparison, we also report results for the 72 languages supported by LaBSE, SONAR, and MEXMA ("o-xsim"), and separately for the 34 languages common to DAP and the other models ("d-xsim"). Results per language are available in Appendix E .

---
[1]https://github.com/facebookresearch/LASER/tree/main/tasks/xsim

| Model | average | SentEval | en | zh | fr | da | nb | pol |
|-------|---------|----------|------|------|------|------|------|------|
| DAP | 61.80 | 78.18 | 66.35 | 67.46 | 63.76 | 52.27 | 51.58 | 53.03 |
| SONAR | 63.02 | 85.82 | 65.63 | 63.13 | 61.88 | 54.01 | 55.59 | 55.09 |
| LaBSE | 62.77 | 85.63 | 66.75 | **68.69** | 62.05 | 49.53 | 50.76 | 56.00 |
| MEXMA | **65.35** | **86.38** | **68.20** | 66.25 | **66.07** | **55.38** | **58.08** | **57.09** |

Table 2: Classification results, reported as accuracy (%), on SentEval and MTEB (last 6 columns), averaged across languages.

| Model | average | en | zh | fr |
|-------|---------|------|------|------|
| DAP | 66.01 | 63.87 | 61.12 | 73.03 |
| SONAR | 69.70 | 70.73 | 60.80 | 77.57 |
| LaBSE | 68.47 | 69.75 | 61.95 | 73.70 |
| MEXMA | **71.55** | **74.39** | **62.12** | **78.13** |

Table 3: Pair classification results, average precision (%), on MTEB, averaged across languages.

The results are shown in Table 1. MEXMA outperforms previous SOTA on all benchmarks, showcasing improved alignment in our approach. The improvements in xsim and BUCC suggest that our approach improves the semantic alignment of the embeddings. The large improvement in xsim++ (+2.48% absolute improvement against the previous best model SONAR) also indicates the increased robustness of our model with regard to hard negatives, likely due to handling better lexical information. For more thorough comparisons using the same data and backbones see Appendix B.4.

## 4.2 DOWNSTREAM TASKS

To understand the quality of our embeddings and how generic they are, we evaluate them on several tasks from the MTEB benchmark (Muennighoff et al., 2023). We report the averaged results for each language. For the full list of results for every task, see Appendix E.

**SINGLE SENTENCE CLASSIFICATION**   We evaluate our model's classification performance on two benchmarks. First, the SentEval suite (Conneau & Kiela, 2018) is used to assess the performance across various tasks in English. We evaluate on the tasks considered in LaBSE. Second, we evaluate the multilingual classification capabilities using the available datasets from the MTEB benchmark. Table 2 shows the aggregated results. We can see that MEXMA outperforms all baseline models on average, and more specifically gains +2.33% when compared with SONAR.

**PAIRWISE SENTENCE CLASSIFICATION**   We further evaluate on the pair classification task. This task consists in classifying sentence pairs, e.g. determining if two sentences are duplicates or not. The metric, as reported in MTEB, is the Average Precision (AP) based on the distance between sentence representations. The results are in Table 3. MEXMA consistently outperforms all baselines on average, by at least +1.85%. These results, combined with our single sentence classification results, suggest that our model can effectively encode the relevant information in the sentence vectors.

**SEMANTIC TEXTUAL SIMILARITY (STS)**   The STS task evaluates the model's ability to replicate human judgments on sentence similarity. The metric, as reported in MTEB, is the Spearman correlation based on distance. The results are in Table 4. We can see that LaBSE outperforms all other methods, and in particular MEXMA by 0.66%. MEXMA outperforms SONAR (+5.95%) and

| Model | avg | eng | zh | fr | pl |
|-------|------|------|------|------|------|
| DAP | 59.39 | 67.45 | 45.31 | 67.74 | 57.06 |
| SONAR | 58.04 | 67.24 | 42.15 | 65.60 | 57.17 |
| LaBSE | **64.65** | **70.93** | 47.50 | **74.33** | **65.82** |
| MEXMA | 63.99 | 70.62 | **51.56** | 70.10 | 63.67 |

Table 4: STS results, reported as Spearman correlation (%), on MTEB, averaged across languages.

| component | xsim ↓ | xsim++ ↓ | SentEval ↑ |
|---|---|---|---|
| Only sentence-level grads ① | 0.15 | 11.37 | 85.06 |
| + Token-level grads ② | 0.10 ↓0.05 | 9.67 ↓1.7 | 85.98 ↑0.92 |
| + KoLeo loss ③ - MEXMA | 0.06 ↓0.04 | 9.60 ↓0.07 | 86.38 ↑0.4 |

Table 5: Ablation study of the different components of the model. All experiments are conducted with the final hyperparameters of the model, as reported in Section 3.1.

| Model | xsim ↓ | xsim++ ↓ | SentEval ↑ |
|---|---|---|---|
| Contrastive XLM-RoBERTa | 0.13 | 33.30 | 85.5 |
| Contrastive MEXMA without MLM token-level gradients | 0.13 | 12.78 | 85.86 |
| Contrastive MEXMA | 0.12 | 10.93 | 85.94 |

Table 6: Using contrastive loss as the alignment loss in MEXMA.

DAP (+4.6%). The results indicate that the contrastive loss better suits the STS task, given that this is the only task where DAP is able to outperform SONAR, and where LaBSE outperforms MEXMA.

## 5 ABLATIONS AND ANALYSES

In this section, we conduct a comprehensive analysis of our MEXMA architecture, examining the impact of its individual components, how it scales with varying model and data sizes, and its potential to improve other alignment approaches. We also examine the characteristics of the token embeddings and sentence representations learned by our model.

### 5.1 MODEL COMPONENTS

In Table 5 we ablate the impact of having direct token-level gradients in MEXMA. The goal is to understand the relevance of the gradients that update the encoder: either only from the sentence, or from the sentence *and* all tokens. In model ①, we have all of MEXMA's components, as covered in Section 3, without the KoLeo loss. However, the gradients from the unmasking task are only back propagating through the sentence representations back to the encoder, and are deactivated for the individual tokens the encoder outputs, i.e. in the $\mathcal{L}_{mlm}$ mentioned in Section 3, $\hat{A}/\hat{B}$ have no gradients flowing back to the encoder. This model already achieves results that are competitive with current state of the art, but does not outperform them. However, if we allow the gradients to flow through the tokens directly, model ②, we are able to outperform the current state-of-the-art. As we hypothesized, adding updates on the tokens directly, coupled with the sentence updates largely improves results across all tasks. Additionally, we also show that adding the KoLeo loss, model ③, also slightly improves results across all tasks. The ablation on all components of the model, and on cross-linguality , is provided in Appendix B.

### 5.2 CONTRASTIVE ALIGNMENT LOSS

To further assess the improvements given by the direct token updates in MEXMA, and understand MEXMA's scalability to other alignment approaches, we replaced our alignment loss, MSE, with a contrastive loss (also dropping the KoLeo loss). We used a siamese network with XLM-RoBERTa-large trained on the symmetric cross-entropy loss (InfoNCE from van den Oord et al. (2019)) as the baseline model, having an architecture similar to LaBSE (Feng et al., 2022). Our training used a batch size of 1.2k, with the rest of the parameters the same as reported in Section 3.1. The results are presented in Table 6. Our baseline model performs well on xsim and SentEval but struggled with xsim++. Switching to the MEXMA architecture without token-level gradients, as done in model ① in Section 5.1, improved performance, already close to state-of-the-art xsim++ performance. Moreover, incorporating token-level gradients, allowing the full MEXMA architecture with contrastive loss, as done in model ② in Section 5.1, resulted in competitive performance, already outperforming previous approaches in SentEval and xsim++. This demonstrates the positive impact of direct

| Model | #parameters | xsim ↓ | xsim++ ↓ | SentEval ↑ | d-xsim ↓ | d-xsim++ ↓ |
|---|---|---|---|---|---|---|
| DAP | 277M | | | 78.18 | 2.90 | 32.82 |
| **MEXMA-base** | 277M | 0.13 | 13.03 | 85.30 | 0.06 | 11.01 |
| LaBSE | 471M | 0.92 | 18.65 | 85.63 | 0.26 | 14.51 |
| **MEXMA** | 559M | **0.06** | **9.60** | **86.38** | **0.02** | **8.26** |
| SONAR | 766M | 0.09 | 12.08 | 85.82 | 0.04 | 10.55 |

Table 7: Model size comparison. MEXMA-base is based on the XLM-RoBERTa-base, and MEXMA is based on XLM-RoBERTa-Large. The d-xxx columns are computed on 34 languages supported by DAP.

| Model | 81 xsim ↓ | 81 xsim++ ↓ | SentEval ↑ | 90 xsim ↓ | 90 xsim++ ↓ | SentEval ↑ |
|---|---|---|---|---|---|---|
| SONAR | 0.09 | 12.08 | 85.82 | **0.05** | 11.42 | 85.82 |
| MEXMA | **0.06** | **9.60** | **86.38** | **0.05** | **9.06** | **86.64** |

Table 8: Training data size comparison. We train MEXMA on either 81 languages, or 90 languages. See Appendix C for the list of covered languages.

token-level gradients and shows that MEXMA can be easily integrated with existing alignment approaches, such as contrastive learning, to improve their results.

## 5.3 MODEL AND DATA SIZES

Table 7 shows how our model's results scale with the model size. We train two models, MEXMA-base with 277M parameters, based on XLM-RoBERTa-base, and MEXMA with 559M parameters, based on XLM-RoBERTa-large. We observe that even the smaller model (277M parameters) outperforms LaBSE (471M parameters), on both xSIM and xSIM++, and gets a close result in SentEval, with a 0.3% decrease in performance, with 58.81% of the size. This smaller model also gets surprisingly close to the results of SONAR, which has 766M parameters, i.e. ≈2.77 times its size. These results show that our approach works on smaller and larger models, and it seems to enable quite powerful small models, due to our stronger training signal. Our larger model, MEXMA, with ≈73% the size of SONAR, is able to largely outperform it across all tasks.

To investigate the impact of training data, we conducted experiments using two different language subsets of the FLORES200. We trained separate MEXMA models on each subset, using the same hyperparameters as reported in Section 3.1. For comparison, we evaluated the publicly available SONAR model, which was trained on all available 200 languages, on both language subsets. The results, presented in Table 8, demonstrate that MEXMA outperforms SONAR on both subsets, highlighting the adaptability and robustness of our approach to varying training data.

## 5.4 MASKING RATIO

NLP models typically use masking percentages around 15%, whereas vision papers have explored much higher masking ratios, ranging from 40% in BEiT (Bao et al., 2022) to as high as 90% in MAE (He et al., 2022) and V-JEPA (Bardes et al., 2024), usually aligning *augmentations*. For text, there is less redundancy and the representations are more information-dense. In our case, we are aligning the same sentence in several languages, which can be viewed as *augmentations* of a pivot sentence, i.e. the sentence in English. We need to know how much we can mask, to make the unmasking task hard, but to not deteriorate the performance of our encoder. More information is provided in Appendix B. The range 30%-60% seems to be the best operating region. We selected 40% for all experiments conducted in this paper, since it had the best balance between alignment and classification.

## 5.5 TOKEN EMBEDDINGS ANALYSIS

Sentence vectors are pooled representations of their tokens. In this section, we investigate the information encoded in the tokens from the last layer across different models. Our goal is to determine whether the tokens primarily convey semantic, lexical, and/or contextual information. Although

| Model | % other | % same language | % same sentence | % translation |
|-------|---------|-----------------|-----------------|---------------|
| XLM-RoBERTa | 1.19 | 63.89 | 2.65 | 32.27 |
| LaBSE | 0.00 | 0.13 | 42.33 | 57.54 |
| DAP | 0.00 | 0.66 | 20.11 | 79.23 |
| No-tok-MEXMA | 0.13 | 0.40 | 11.90 | 87.57 |
| NLLB | 0.40 | 3.17 | 1.72 | 94.71 |
| SONAR | 0.00 | 0.13 | 0.20 | 99.67 |
| MEXMA | 0.26 | 1.33 | 0.53 | 97.88 |

Table 9: Result of the token matching analysis.

these categories can be intertwined, understanding the dominant characteristics of each model's tokens provides valuable insights into their behavior.

To gain insight into the information encoded in individual tokens, we examined their nearest neighbors in the embedding space. We categorized these neighboring tokens into four groups based on the sentence they belong to. *Same language*: the matched token is the same token in a different sentence in the same language, which means that it encodes lexical information. *Same sentence*: the token matches another one in the same sentence, meaning the tokens' representations are heavily influenced by its context. *Translation*: the token matches its equivalent in a translation of the original sentence. It means that the tokens' representations are aligned across languages. *Other*: tokens that do not belong to previous classes.

We conducted these experiments by encoding all tokens from all sentences of the 81 languages (see Appendix C for the list) on the FLORES200 test set using each model. We randomly select three tokens among each of the first 250 English sentences of the dataset as query, and for each query, we retrieve the five closest tokens among all tokens of all sentences (but itself). We analyze the properties of the sentence encoders as well as some respective backbones, XLM-RoBERTa (used to initialize MEXMA) and NLLB-200 encoder (used in SONAR). For the sake of comparison, we also examine "no-tok-MEXMA", a variant of MEXMA that does not use token-level gradients during training. The statistics are shown in Table 9.

Our analysis reveals distinct characteristics for the considered models and we can cluster them in three different overall behaviours. XLM-RoBERTa exhibits strong lexical relationships (high *same language* percentage) but weaker semantic and contextual relations.

LaBSE, DAP and no-tok-MEXMA show higher semantic capabilities as shown by the larger *translation* rate. However, we can also observe a high percentage of matches with adjacent tokens (*same sentence* column), indicating that those models encode a very large amount of context in their tokens.

NLLB, SONAR and MEXMA have strong cross-lingual semantic capability as shown by the very high percentage in the *translation* column. This is expected as SONAR and NLLB were trained to perform translation, and MEXMA cross-lingual unmasking. Notice that for SONAR and MEXMA, this cross-lingual token level alignment is guided by the decoding using the sentence representation as context (and additionally the direct token-level gradients for MEXMA).

Note also that LaBSE and DAP are the only models trained with a sentence-level contrastive loss, and even though DAP has an additional loss to enforce the semantic alignment of the tokens, it does not manage to achieve the same alignment as SONAR and MEXMA.

Notably, comparing the backbones NLLB and XLM-RoBERTa, we can see that the former exhibits more semantical tokens than the latter, as shown by its higher *translation* rate and lower *same sentence* rate, which can be attributed to its translation-based pre-training that enhances semantic properties and cross-lingual alignment. SONAR, which starts from NLLB, also matches translated tokens with a high rate, >99%, but does not encode a lot of lexical information (low *same language* rate). MEXMA also matches translated tokens very frequently, but additionally displays more lexicality (higher *same language* rate) and increased semantic robustness (higher *other* rate). To assess the latter, we verified MEXMA's *other* matches. The matched tokens belong to sentences in other languages that are not translations of the original one, with the matched token being the translated token. We believe that MEXMA produces sentence representations that inherit the above properties,

| Model | xsim ↓ | xsim++ ↓ | STS ↑ | Classification ↑ |
|---|---|---|---|---|
| Uni LaBSE | 2.02 | 20.73 | 63.50 | 58.03 |
| Uni MEXMA | 0.19 | 18.21 | 54.24 | 56.98 |
| CLS LaBSE | 0.92 | 18.65 | 64.65 | 62.77 |
| CLS MEXMA | 0.06 | 9.60 | 63.99 | 65.35 |
| Δ LaBSE | -119.65 | -11.19 | +1.78 | +7.55 |
| Δ MEXMA | -212.50 | -89.73 | +15.24 | +12.81 |

Table 10: Downstream results for LaBSE and MEXMA, using both a uniform attention distribution (Uni xxx in the table), and the CLS distribution (CLS xxx in the table). The last two rows provide the delta between the uniform and CLS distributions, in relative terms. Classification and STS results are across all datasets mentioned under Appendix E.

allowing it to outperform other models on downstream tasks. We provide examples to illustrate the behavior of the models, also experiments with SimAlign (Jalili Sabet et al., 2020) , in Appendix F.

### 5.6 SENTENCE VECTOR ANALYSIS

Sentence representations are created by combining token representations in various ways (average or CLS/attention pooling). The previous section examined properties encoded in tokens, and this section explores how these representations are combined to form the sentence embedding.

For SONAR, the attention weight distribution is uniform, given that SONAR averages the tokens to create their sentence representation. MEXMA and LaBSE both use a CLS token to perform pooling over the tokens in the sentence.

MEXMA's and LaBSE's attention distribution are rather different, with LaBSE having a more uniform attention distribution across its tokens, and MEXMA having a more skewed representation. We verify this by computing the average entropy of the attention probabilities in the last layer given by the CLS token, for both models on the test set of the FLORES200, in the languages supported by both LaBSE and MEXMA. LaBSE gets an entropy of $\approx 3.4$, while MEXMA gets an entropy of $\approx 2.5$. The entropy values obtained for LaBSE and MEXMA are difficult to interpret in absolute terms, but the relative difference between them is informative. Specifically, LaBSE exhibits a higher entropy compared to MEXMA, suggesting that it has a more uniform distribution of attention probabilities. We provide examples of the distributions in Appendix G.

We perform an additional analysis, where we push the uniformity of the sentence representation to the extreme, by using the average of tokens as our sentence representation. By doing this for both MEXMA and LaBSE, we aim to understand the importance/impact of the attention distribution for each model. The results are provided in Table 10. The deltas are computed in terms of relative change from the uniform to the CLS representation. We can see that for all tasks, MEXMA has a larger change in performance compared to LaBSE, showing that indeed since our representations are more skewed, we suffer more from an increase in uniformity of the distribution. For those tasks, it is noticeable that MEXMA having a uniform distribution, will lose its ability to focus on the important tokens, decreasing its results. For LaBSE the decrease is not as accentuated, since it was already not focusing as much on the important tokens with its more uniform CLS pooling.

## 6 CONCLUSION

We introduced MEXMA, a novel multilingual alignment technique that leverages both token-level and sentence-level objectives. We show that integrating token-level objectives into the training of cross-lingual sentence encoders (CLSE) greatly improves sentence representation quality, outperforming current state-of-the-art pre-trained CLSE in bitext mining and other downstream tasks. We additionally validate these improvements via ablations. Notably, MEXMA also achieves strong token alignment across languages and effectively encodes meaningful information within each token. Since the sentence representation is built from these tokens, as we analysed, this leads to better sentence representations. Looking ahead, we plan to explore MEXMA's scalability to more languages, and potentially modalities.

## 7 REPRODUCIBILITY STATEMENT

In order to ensure reproducibility of our results we detail the hyperparameters used to train our network in Section 3.1 and Appendix A. Additionally, we provide the training code as supplementary material, and will publicly release the code and the model weights after the paper is no longer anonymous.

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

# A   EXPERIMENTAL SETUP

## A.1   ENCODER BACKBONE

The available implementation of XLM-RoBERTa in HuggingFace employs an inefficient attention mechanism, which we have modified to incorporate the memory-efficient attention from xFormers (Lefaudeux et al., 2022). This modification was necessary due to the random batching process used in our training, which results in a significant amount of padding and increased computational cost. To address this issue and eliminate padding, we have employed the BlockDiagonalMask [2], which through custom CUDA kernels, avoids computations in padding altogether. With this change we are able to increase our batch size in each GPU by a factor of $\approx 8$.

## A.2   UNMASKING HEAD

For the unmasking head, we use 6 transformer layers, also leveraging the memory-efficient attention.

## A.3   COMPUTE AND TRAINING LENGTH

Our models were trained on a single node of 8 A100 GPUs. Each GPU had a batch size of 150, totalling 1,200 batch size across all GPUs. We accumulated two gradients, making our effective batch size 2,400. We trained our models for 300k steps.

## A.4   LOSSES

Our models were trained with $\alpha = 1$, $\beta = \frac{1}{2}$ and $\gamma = \frac{0.01}{2}$.

## A.5   TRAINING PARAMETERS

We utilize the AdamW optimizer for our training process. The learning rate is linearly increased from 1e-5 for the 300k steps. To optimize memory usage, we employ mixed precision training, where the model is stored in float32, while most computations are performed in float16. The maximum sequence length for our input data is set to 200 tokens. Finally, we apply a masking ratio of 40% to the input data.

---

[2]`https://facebookresearch.github.io/xformers/components/ops.html#xformers.ops.fmha.attn_bias.BlockDiagonalMask`

# B ABLATIONS

## B.1 MODEL COMPONENTS

| component | xsim ↓ | xsim++ ↓ | SentEval ↑ |
|---|---|---|---|
| Non-symmetrical ① | 0.09 | 14.75 | 84.68 |
| + Symmetrical architecture ② | 0.09 ₀.₀₀ | 14.39 ↓0.36 | 84.83 ↑0.15 |
| + Alignment loss (clean to dirty alignment) ③ | 0.21 ↑0.12 | 12.09 ↓2.3 | 85.61 ↑0.78 |
| + Clean to clean alignment ④ | 0.15 ↓0.06 | 11.37 ↓0.72 | 85.06 ↓0.55 |
| + Token-level grads ⑤ | 0.10 ↓0.05 | 9.67 ↓1.7 | 85.98 ↑0.92 |
| + KoLeo loss ⑥ - MEXMA | 0.06 ↓0.04 | 9.60 ↓0.07 | 86.38 ↑0.4 |

Table 11: Ablation study of the different components of the model. All experiments are conducted with the final parameters of the model, as reported in Section 3.1.

In Table 11, we ablate the different components of our architecture described in Section 3. We briefly explain each entry in the table. Model ① has only two encoder instances, one for each language, where one of the inputs is masked, and the other is left clean. The token unmasking is performed with the clean sentence representation as context. The languages are randomly swapped at every new sample, to eliminate potential biases. The gradients from the unmasking task are only propagated back to the encoder via the sentence representation, and there is no gradient propagation from the individual tokens back to the encoder. There is also neither alignment nor koleo losses. Model ② adds two additional encoder instances, totalling four instances, two for each language, where now each language has its clean and masked input. This allows to unmask language $\mathcal{A}$ with language $\mathcal{B}$, and vice-versa, and will also allow (once added) to align two clean sentence representations. Model ③ adds the alignment loss, but it is performed between the masked sentence representation of language $\mathcal{A}$ and the clean sentence representation of language $\mathcal{B}$, to better emphasize the advantages of having a symmetrical architecture with an alignment loss between two clean representations. Model ④ then changes the alignment loss to be performed between the two clean sentence representations of each language. In model ⑤ we allow the gradients from the unmasking to be propagated to the encoder via each individual token, as well as its sentence representation. Finally, model ⑥ adds the KoLeo loss.

The results indicate that each component always enhances performance on at least two out of the three tasks. Notably, the alignment loss, ③-④, and token-level gradients, ⑤, emerge as the most critical components. More precisely, the alignment loss yields substantial improvements on two tasks while also resulting in a notable decline in performance on another task. In contrast, the token-level gradients consistently provide significant performance gains across all three tasks.

## B.2 CROSS-LINGUALITY

| component | xsim ↓ | xsim++ ↓ | SentEval ↑ |
|---|---|---|---|
| Same language unmasking | 21.83 | 73.78 | 80.34 |
| Cross lingual unmasking | 0.06 ↓21.77 | 9.60 ↓64.18 | 86.38 ↑6.04 |

Table 12: Ablation study of the importance of cross-lingual unmasking. All experiments are conducted with the final parameters of the model, as reported in Section 3.1.

In Table 12 we ablate the importance of cross-linguality in the unmasking. To conduct this experiment, we performed the unmasking using as context the sentence representation in the same language of the tokens being unmasked, instead of the representation in the opposite language. The large gap in the results shows the importance of doing the unmasking cross-lingually, as motivated in Section 3. The experiments were conducted using the same hyperparameters.

| Masking % | xsim ↓ | xsim++ ↓ | SentEval ↑ |
|---|---|---|---|
| 20% | 0.06 | 10.50 | 85.87 |
| 30% | 0.06 | 9.82 | 86.00 |
| 40% | 0.06 | 9.60 | 86.38 |
| 50% | 0.07 | 9.56 | 86.37 |
| 60% | 0.08 | 9.79 | 86.13 |
| 70% | 0.09 | 10.65 | 86.41 |
| 80% | 0.10 | 12.81 | 85.85 |
| 90% | 0.11 | 14.62 | 84.99 |

Table 13: The model performance across different masking ratios.

### B.3 MASKING RATIO

Classical NLP masked encoders like BERT use a small masking percentage, usually $\approx 15\%$, without aligning any *augmentations*. Recent vision approaches use much higher masking percentages. BEiT (Bao et al., 2022) was one of the first masked image modelling (MIM) approaches, in a BERT-style training, and masked 40%. MAE (He et al., 2022) is another BERT-like model for images, and masks 75%, but shows that even masking 80% or 90% still achieves good results. DINO v2 (Oquab et al., 2024) and I-BOT (Zhou et al., 2022) mask between 10%-50% in a block-wise masking scenario, aligning *augmentations*. I-BOT can use 65%-75% masking ratio, if randomly masking (instead of block-wise masking). For videos, V-JEPA (Bardes et al., 2024) masks with a very high percentage of 90%. Recent textual approaches, namely RetroMAE experiment with masking percentages of up to 50~70%, but this task will not update the actual encoder.

For MEXMA, since these masking gradients are updating our encoder, we need to strive for a balance where unmasking is hard, and cannot be done by the encoder and head, but also not too much that will deteriorate the representations of the encoder. Table 13 shows the results we obtained for the different masking ratios.

### B.4 FAIRER COMPARISONS

| Model | xsim ↓ | xsim++ ↓ | SentEval ↑ |
|---|---|---|---|
| XLM-R LaBSE | 0.10 | 33.82 | 86.08 |
| NLLB-MEXMA | 0.11 | 23.36 | 85.20 |
| Baselines | | | |
| MEXMA | 0.06 | 9.60 | 86.36 |
| SONAR | 0.09 | 12.08 | 85.82 |
| LaBSE | 0.92 | 18.65 | 85.63 |

Table 14: Fairer comparisons with same backbone to compare different strategies, all trained on the same data.

In this section, we conduct additional experiments to enhance the fairness of our method comparisons, ensuring that all models are trained on the same data as MEXMA. Results are reported in Table 14. To facilitate a more equitable comparison with LaBSE, we re-implemented LaBSE using the same backbone as MEXMA, i.e. XLM-RoBERTa (XLM-R). The model was trained with the same hyperparameters reported in the original paper, specifically a 4k batch size (compared to the 1k batch size used in MEXMA). This different backbone and data led to improved performance over LaBSE on the xsim and SentEval benchmarks, although it resulted in a significant decrease in performance on the xsim++ benchmark. For a more accurate comparison with SONAR, we replaced the XLM-R backbone in MEXMA with the NLLB encoder. This approach is more straightforward than training SONAR with XLM-R, as SONAR requires a pre-trained translation decoder. The results, however, were inferior to those of MEXMA across all tasks, with larger gaps than SONAR. This outcome is expected, given that the NLLB model was not originally trained for masked language modeling.

## C LANGUAGE INFORMATION APPENDIX

In this section, we cover the languages used by our model. The list of languages used to train our model is reported in Table 15. The list used to conduct the experiments with 90 languages is available in Table 16.

| FLORES200 code | Language | FLORES200 code | Language |
|---|---|---|---|
| acm_Arab | Mesopotamian Arabic | kan_Knda | Kannada |
| aeb_Arab | Tunisian Arabic | kat_Geor | Georgian |
| afr_Latn | Afrikaans | kaz_Cyrl | Kazakh |
| amh_Ethi | Amharic | khm_Khmr | Khmer |
| ary_Arab | Moroccan Arabic | kir_Cyrl | Kyrgyz |
| arz_Arab | Egyptian Arabic | kor_Hang | Korean |
| asm_Beng | Assamese | lao_Laoo | Lao |
| azb_Arab | South Azerbaijani | mal_Mlym | Malayalam |
| azj_Latn | Azerbaijani | mar_Deva | Marathi |
| bel_Cyrl | Belarusian | mkd_Cyrl | Macedonian |
| ben_Beng | Bengali | mya_Mymr | Burmese |
| bos_Latn | Bosnian | nld_Latn | Dutch |
| bul_Cyrl | Bulgarian | nno_Latn | Norwegian |
| cat_Latn | Catalan | nob_Latn | Norwegian Bokmål |
| ces_Latn | Czech | npi_Deva | Nepali |
| ckb_Arab | Central Kurdish | pol_Latn | Polish |
| cym_Latn | Welsh | por_Latn | Portuguese |
| dan_Latn | Danish | ron_Latn | Romanian |
| deu_Latn | German | rus_Cyrl | Russian |
| ell_Grek | Greek | san_Deva | Sanskrit |
| eng_Latn | English | sin_Sinh | Sinhala |
| epo_Latn | Esperanto | slk_Latn | Slovak |
| est_Latn | Estonian | slv_Latn | Slovenian |
| eus_Latn | Basque | snd_Arab | Sindhi |
| fin_Latn | Finnish | som_Latn | Somali |
| fra_Latn | French | spa_Latn | Spanish |
| gla_Latn | Scottish Gaelic | srp_Cyrl | Serbian |
| gle_Latn | Irish | sun_Latn | Sundanese |
| glg_Latn | Galician | swe_Latn | Swedish |
| guj_Gujr | Gujarati | swh_Latn | Swahili |
| hau_Latn | Hausa | tam_Taml | Tamil |
| heb_Hebr | Hebrew | tel_Telu | Telugu |
| hin_Deva | Hindi | tha_Thai | Thai |
| hrv_Latn | Croatian | tur_Latn | Turkish |
| hun_Latn | Hungarian | uig_Arab | Uyghur |
| hye_Armn | Armenian | ukr_Cyrl | Ukrainian |
| ind_Latn | Indonesian | urd_Arab | Urdu |
| isl_Latn | Icelandic | vie_Latn | Vietnamese |
| ita_Latn | Italian | xho_Latn | Xhosa |
| jav_Latn | Javanese | zho_Hant | Chinese (Traditional) |
| jpn_Jpan | Japanese | | |

Table 15: 81 languages set.

| FLORES200 code | Language | FLORES200 code | Language |
|---|---|---|---|
| afr_Latn | Afrikaans | kmr_Latn | Kurdish (Kurmanji) |
| als_Latn | Albanian | kor_Hang | Korean |
| amh_Ethi | Amharic | lao_Laoo | Lao |
| arb_Arab | Arabic | lit_Latn | Lithuanian |
| asm_Beng | Assamese | lvs_Latn | Latvian |
| azj_Latn | Azerbaijani | mal_Mlym | Malayalam |
| bel_Cyrl | Belarusian | mar_Deva | Marathi |
| ben_Beng | Bengali | mkd_Cyrl | Macedonian |
| bos_Latn | Bosnian | mya_Mymr | Burmese |
| bul_Cyrl | Bulgarian | nld_Latn | Dutch |
| cat_Latn | Catalan | nno_Latn | Norwegian |
| ces_Latn | Czech | npi_Deva | Nepali |
| cym_Latn | Welsh | ory_Orya | Oriya |
| dan_Latn | Danish | pan_Guru | Punjabi |
| deu_Latn | German | pbt_Arab | Pashto |
| ell_Grek | Greek | plt_Latn | Malagasy |
| eng_Latn | English | pol_Latn | Polish |
| epo_Latn | Esperanto | por_Latn | Portuguese |
| est_Latn | Estonian | prs_Arab | Persian |
| eus_Latn | Basque | ron_Latn | Romanian |
| fin_Latn | Finnish | rus_Cyrl | Russian |
| fra_Latn | French | san_Deva | Sanskrit |
| gaz_Latn | Oromo | sin_Sinh | Sinhala |
| gla_Latn | Gaelic | slk_Latn | Slovak |
| gle_Latn | Irish | slv_Latn | Slovenian |
| glg_Latn | Galician | snd_Arab | Sindhi |
| guj_Gujr | Gujarati | som_Latn | Somali |
| hau_Latn | Hausa | spa_Latn | Spanish |
| heb_Hebr | Hebrew | srp_Cyrl | Serbian |
| hin_Deva | Hindi | sun_Latn | Sundanese |
| hrv_Latn | Croatian | swe_Latn | Swedish |
| hun_Latn | Hungarian | swh_Latn | Swahili |
| hye_Armn | Armenian | tam_Taml | Tamil |
| ind_Latn | Indonesian | tel_Telu | Telugu |
| isl_Latn | Icelandic | tha_Thai | Thai |
| ita_Latn | Italian | tur_Latn | Turkish |
| jav_Latn | Javanese | uig_Arab | Uyghur |
| jpn_Jpan | Japanese | ukr_Cyrl | Ukrainian |
| kan_Knda | Kannada | urd_Arab | Urdu |
| kat_Geor | Georgian | uzn_Latn | Uzbek |
| kaz_Cyrl | Kazakh | vie_Latn | Vietnamese |
| khk_Cyrl | Mongolian | xho_Latn | Xhosa |
| khm_Khmr | Khmer | ydd_Hebr | Yiddish |
| kir_Cyrl | Kyrgyz | zho_Hans | Chinese (Simplified) |
| zsm_Latn | Malay | zho_Hant | Chinese (Traditional) |

Table 16: 90 languages set

# D DATASETS

In this section we report the data used to train our models. Table 17 reports all the datasets used to train the models.

| Dataset | Source | Origin |
|---------|--------|--------|
| bible-uedin | Opus | Christodouloupoulos & Steedman (2015); Tiedemann (2012) |
| DGT | Opus | Steinberger et al. (2012); Tiedemann (2012) |
| ECB | Opus | Tiedemann (2012) |
| EMEA | Opus | Tiedemann (2012) |
| EUbookshop | Opus | Tiedemann (2012) |
| infopankki | Opus | Tiedemann (2012) |
| memat | Opus | Tiedemann (2012) |
| OpenSubtitles | Opus | Lison & Tiedemann (2016); Tiedemann (2012), Link: opensubtitles.org |
| QED | Opus | Abdelali et al. (2014); Tiedemann (2012) |
| Tanzil | Opus | Tiedemann (2012), Link: tanzil.net/trans |
| Tatoeba | Opus | Tiedemann (2012) |
| Ted20 | Opus | Reimers & Gurevych (2020); Tiedemann (2012) |
| Tico19 | Opus | Anastasopoulos et al. (2020); Tiedemann (2012) |
| UNPC | Opus | Ziemski et al. (2016); Tiedemann (2012) |
| Wikimedia | Opus | Tiedemann (2012) |
| NLLB mined | Opus | Schwenk et al. (2020); Fan et al. (2020); Tiedemann (2012) |

Table 17: Datasets used to train our models.

# E  MTEB DATASETS

In this section, we report the scores for each task of the MTEB benchmark reported in Section 4. We report the scores per task, with every dataset used per task, and per language. MEXMA is able to outperform the previous SOTA results on mining, while also improving the downstream results on classification and pair classification. LaBSE outperforms all other models on STS.

## E.1  BITEXT MINING

Results for mining are in Table 18, for the BUCC dataset. We report the scores on the four available languages: German, French, Russian and Chinese. Results on all languages covered by MEXMA for xsim and xsim++ on FLORES200 are provided in Tables 19 and 20, respectively.

| LP | DAP | SONAR | LaBSE | MEXMA |
|---|---|---|---|---|
| de-en | 99.45 | 98.82 | 99.35 | 99.52 |
| fr-en | 98.58 | 98.09 | 98.72 | 98.98 |
| ru-en | 97.74 | 97.37 | 97.78 | 98.06 |
| zh-en | 98.96 | 98.72 | 99.16 | 99.18 |

Table 18: BUCC results for each language pair (LP).

## E.2  CLASSIFICATION

Classification results for English are available in Table 21, for SentEval, and in Table 22 for the English MTEB classification datasets. Classification results for Chinese, French, Danish, Norwegian and Polish are reported in Table 23, Table 24, Table 25, Table 26, Table 27, respectively. MEXMA outperforms all other models on average.

## E.3  PAIR CLASSIFICATION

Pair classification results for English, French and Chinese are reported in Table 28, Table 29, and Table 30, respectively. MEXMA outperforms all other models on average.

## E.4  SEMANTIC TEXTUAL SIMILARITY (STS)

Semantic Textual Similarity (STS) results are reported in Table 31, Table 33, Table 34 and Table 32 for English, French, Polish and Chinese, respectively. LaBSE outperforms MEXMA and the remaining models on STS. MEXMA and LaBSE outperform SONAR by large margins.

| Language | SONAR | LaBSE | MEXMA | DAP | Language | SONAR | LaBSE | MEXMA | DAP |
|----------|-------|-------|-------|-----|----------|-------|-------|-------|-----|
| acm_Arab | 0 | 0 | 0 | - | kan_Knda | 0 | 0 | 0 | - |
| aeb_Arab | 0.10 | 0.10 | 0.10 | - | kat_Geor | 0.40 | 0 | 0 | 7.41 |
| afr_Latn | 0 | 0 | 0 | 0.10 | kaz_Cyrl | 0.20 | 0.20 | 0.20 | 44.96 |
| amh_Ethi | 0 | 0 | 0 | - | khm_Khmr | 0 | 2.08 | 0 | - |
| ary_Arab | 0.79 | 1.09 | 0.89 | - | kir_Cyrl | 0.10 | 0 | 0 | - |
| arz_Arab | 0 | 0 | 0 | - | kor_Hang | 0 | 0 | 0 | 0 |
| asm_Beng | 0 | 0 | 0 | - | lao_Laoo | 0 | 2.77 | 0 | 0.20 |
| azb_Arab | 1.68 | 9.58 | 0.99 | - | mal_Mlym | 0.10 | 0.10 | 0.10 | 1.48 |
| azj_Latn | 0.20 | 0.10 | 0.10 | - | mar_Deva | 0 | 0 | 0 | 1.38 |
| bel_Cyrl | 0.30 | 0 | 0 | - | mkd_Cyrl | 0 | 0 | 0 | - |
| ben_Beng | 0 | 0 | 0 | 0 | mya_Mymr | 0.20 | 0.30 | 0.20 | - |
| bos_Latn | 0 | 0 | 0 | - | nld_Latn | 0.10 | 0 | 0 | 0 |
| bul_Cyrl | 0.10 | 0 | 0 | 0 | nno_Latn | 0.10 | 0 | 0.10 | - |
| cat_Latn | 0 | 0 | 0 | - | nob_Latn | 0.10 | 0.10 | 0.10 | - |
| ces_Latn | 0 | 0 | 0 | - | npi_Deva | 0.40 | 0.30 | 0.30 | - |
| ckb_Arab | 0.10 | 49.11 | 0 | - | pol_Latn | 0 | 0 | 0 | - |
| cym_Latn | 0 | 0 | 0 | - | por_Latn | 0 | 0 | 0 | 0 |
| dan_Latn | 0 | 0 | 0 | - | ron_Latn | 0 | 0 | 0 | - |
| deu_Latn | 0 | 0 | 0 | 0 | rus_Cyrl | 0.10 | 0 | 0 | 0 |
| ell_Grek | 0 | 0 | 0 | 0.10 | san_Deva | 0.50 | 0.79 | 0.40 | - |
| epo_Latn | 0 | 0 | 0 | - | sin_Sinh | 0 | 0 | 0 | - |
| est_Latn | 0 | 0 | 0 | 0 | slk_Latn | 0 | 0 | 0 | - |
| eus_Latn | 0 | 0 | 0 | 0 | slv_Latn | 0.10 | 0 | 0 | - |
| fin_Latn | 0.10 | 0.10 | 0.10 | 0.10 | snd_Arab | 0 | 0 | 0 | - |
| fra_Latn | 0 | 0 | 0 | 0 | som_Latn | 0.10 | 0.20 | 0.10 | - |
| gla_Latn | 0.10 | 0.10 | 0.10 | - | spa_Latn | 0.10 | 0.10 | 0.10 | 0.10 |
| gle_Latn | 0 | 0 | 0 | - | srp_Cyrl | 0 | 0 | 0 | - |
| glg_Latn | 0 | 0 | 0 | - | sun_Latn | 0.10 | 0.10 | 0.10 | - |
| guj_Gujr | 0 | 0 | 0 | - | swe_Latn | 0 | 0 | 0 | - |
| hau_Latn | 0.30 | 0.30 | 0.30 | - | swh_Latn | 0 | 0 | 0 | 0 |
| heb_Hebr | 0 | 0 | 0 | 0 | tam_Taml | 0 | 0 | 0 | 28.26 |
| hin_Deva | 0.10 | 0 | 0 | 0.10 | tel_Telu | 0 | 0 | 0 | 2.77 |
| hrv_Latn | 0 | 0 | 0 | - | tha_Thai | 0 | 5.53 | 0.10 | 0.10 |
| hun_Latn | 0 | 0 | 0 | 0 | tur_Latn | 0 | 0 | 0 | 0 |
| hye_Armn | 0 | 0 | 0 | - | uig_Arab | 0.10 | 0.10 | 0.10 | - |
| ind_Latn | 0 | 0 | 0 | 0 | ukr_Cyrl | 0 | 0 | 0 | - |
| isl_Latn | 0.20 | 0.10 | 0.10 | - | urd_Arab | 0.10 | 0.10 | 0.10 | 0.30 |
| ita_Latn | 0 | 0 | 0 | 0 | vie_Latn | 0 | 0 | 0 | 0 |
| jav_Latn | 0 | 0 | 0 | 11.17 | xho_Latn | 0.10 | 0.10 | 0.10 | - |
| jpn_Jpan | 0 | 0 | 0 | 0 | zho_Hant | 0.10 | 0 | 0 | 0 |

Table 19: xsim results for each language in FLORES200 covered by MEXMA.

| Language | SONAR | LaBSE | MEXMA | DAP | Language | SONAR | LaBSE | MEXMA | DAP |
|----------|-------|-------|-------|-----|----------|-------|-------|-------|-----|
| acm_Arab | 13.54 | 28.56 | 12.35 | - | kan_Knda | 16.21 | 18.38 | 10.77 | - |
| aeb_Arab | 14.23 | 35.38 | 14.82 | - | kat_Geor | 16.01 | 18.48 | 11.66 | 69.66 |
| afr_Latn | 6.62 | 9.39 | 5.63 | 20.75 | kaz_Cyrl | 12.55 | 15.32 | 8.89 | 89.72 |
| amh_Ethi | 11.56 | 19.07 | 7.51 | - | khm_Khmr | 14.72 | 20.55 | 9.39 | - |
| ary_Arab | 15.91 | 44.47 | 25.59 | - | kir_Cyrl | 15.12 | 20.55 | 13.04 | - |
| arz_Arab | 13.93 | 31.03 | 13.24 | - | kor_Hang | 14.82 | 18.58 | 9.19 | - |
| asm_Beng | 17.98 | 41.11 | 13.44 | - | lao_Laoo | 10.18 | 18.77 | 7.41 | 42.19 |
| azb_Arab | 45.26 | 69.17 | 33.00 | - | mal_Mlym | 13.14 | 19.96 | 11.17 | 54.35 |
| azj_Latn | 17.69 | 17.69 | 12.35 | - | mar_Deva | 10.97 | 15.42 | 8.00 | 54.45 |
| bel_Cyrl | 20.26 | 21.94 | 13.44 | - | mkd_Cyrl | 7.51 | 11.86 | 6.42 | - |
| ben_Beng | 13.83 | 17.79 | 8.70 | 33.60 | mya_Mymr | 19.66 | 28.06 | 15.91 | - |
| bos_Latn | 7.61 | 8.10 | 5.24 | - | nld_Latn | 13.34 | 13.34 | 10.08 | 20.45 |
| bul_Cyrl | 9.19 | 9.19 | 5.53 | 17.89 | nno_Latn | 16.80 | 13.24 | 8.30 | - |
| cat_Latn | 6.03 | 8.79 | 5.04 | - | nob_Latn | 15.51 | 11.56 | 7.41 | - |
| ces_Latn | 8.20 | 11.76 | 6.72 | - | npi_Deva | 14.53 | 13.74 | 7.61 | - |
| ckb_Arab | 13.64 | 93.97 | 14.03 | - | pol_Latn | 11.17 | 12.65 | 8.70 | - |
| cym_Latn | 7.61 | 14.03 | 5.43 | - | por_Latn | 5.93 | 9.09 | 6.32 | 14.53 |
| dan_Latn | 6.03 | 8.10 | 4.84 | - | ron_Latn | 8.10 | 8.40 | 5.73 | - |
| deu_Latn | 6.13 | 7.61 | 6.13 | 15.22 | rus_Cyrl | 7.91 | 9.98 | 6.23 | 19.17 |
| ell_Grek | 10.57 | 16.40 | 8.99 | 26.58 | san_Deva | 24.41 | 51.09 | 22.33 | - |
| epo_Latn | 6.13 | 9.49 | 5.63 | - | sin_Sinh | 12.15 | 16.01 | 7.91 | - |
| est_Latn | 8.10 | 11.46 | 5.93 | 18.87 | slk_Latn | 8.99 | 10.77 | 7.51 | - |
| eus_Latn | 10.87 | 15.32 | 8.30 | 25.20 | slv_Latn | 9.58 | 11.56 | 6.62 | - |
| fin_Latn | 8.99 | 13.44 | 8.50 | 20.55 | snd_Arab | 13.64 | 28.85 | 9.68 | - |
| fra_Latn | 5.93 | 7.61 | 5.34 | 17.59 | som_Latn | 15.81 | 30.93 | 14.92 | - |
| gla_Latn | 17.19 | 23.62 | 12.25 | - | spa_Latn | 9.49 | 11.07 | 7.71 | 20.55 |
| gle_Latn | 10.57 | 15.81 | 9.68 | - | srp_Cyrl | 6.92 | 9.98 | 5.34 | - |
| glg_Latn | 7.51 | 8.40 | 5.63 | - | sun_Latn | 15.02 | 16.50 | 10.38 | - |
| guj_Gujr | 11.56 | 15.12 | 8.30 | - | swe_Latn | 8.00 | 8.99 | 6.03 | - |
| hau_Latn | 16.40 | 25.99 | 13.44 | - | swh_Latn | 7.11 | 15.71 | 8.89 | 29.05 |
| heb_Hebr | 6.92 | 15.02 | 7.51 | 26.28 | tam_Taml | 15.61 | 18.48 | 11.26 | 81.32 |
| hin_Deva | 9.58 | 10.97 | 6.92 | 29.74 | tel_Telu | 13.83 | 15.12 | 10.87 | 57.02 |
| hrv_Latn | 8.20 | 9.09 | 6.52 | - | tha_Thai | 10.57 | 28.16 | 8.20 | 30.83 |
| hun_Latn | 9.09 | 13.74 | 7.91 | 17.79 | tur_Latn | 8.60 | 10.87 | 7.51 | 18.38 |
| hye_Armn | 7.51 | 12.94 | 9.09 | - | uig_Arab | 16.70 | 23.12 | 13.74 | - |
| ind_Latn | 6.23 | 9.09 | 5.73 | 14.92 | ukr_Cyrl | 10.08 | 12.25 | 7.61 | - |
| isl_Latn | 10.38 | 14.43 | 8.50 | - | urd_Arab | 12.25 | 16.70 | 10.08 | 47.13 |
| ita_Latn | 9.98 | 9.49 | 6.23 | 16.11 | vie_Latn | 7.41 | 12.15 | 7.61 | 18.58 |
| jav_Latn | 13.74 | 17.09 | 9.88 | 63.04 | xho_Latn | 11.96 | 31.42 | 15.61 | - |
| jpn_Jpan | 15.22 | 17.79 | 10.08 | 27.17 | zho_Hant | 17.89 | 24.60 | 12.55 | 28.56 |

Table 20: xsim++ results for each language in FLORES200 covered by MEXMA.

| Task | DAP | SONAR | LaBSE | MEXMA |
|------|-----|-------|-------|-------|
| Average | 78.18 | 85.82 | 85.63 | 86.38 |
| MR | 74.33 | 81.23 | 78.89 | 80.14 |
| SST2 | 81.88 | 86.49 | 83.64 | 86.16 |
| TREC | 75.00 | 95.00 | 92.80 | 94.80 |
| CR | 78.70 | 85.67 | 86.44 | 84.43 |
| SUBJ | 91.83 | 93.70 | 93.14 | 94.27 |
| MPQA | 78.86 | 89.38 | 89.66 | 89.41 |
| MRPC | 66.67 | 69.28 | 74.84 | 75.42 |

Table 21: SentEval results.

| Dataset | DAP | SONAR | LaBSE | MEXMA |
|---|---|---|---|---|
| Average | 66.35 | 65.63 | 66.75 | 68.20 |
| AmazonCounterfactualClassification | 77.16 | 81.49 | 75.93 | 78.06 |
| AmazonPolarityClassification | 65.73 | 62.73 | 68.95 | 64.96 |
| AmazonReviewsClassification | 34.03 | 31.55 | 35.80 | 32.77 |
| Banking77Classification | 71.83 | 73.50 | 69.85 | 75.14 |
| ImdbClassification | 62.06 | 55.75 | 62.04 | 62.08 |
| MTOPDomainClassification | 85.54 | 89.92 | 86.06 | 89.85 |
| MTOPIntentClassification | 64.17 | 70.85 | 63.03 | 75.18 |
| MasakhaNEWSClassification | 77.95 | 55.42 | 77.77 | 72.28 |
| MassiveIntentClassification | 63.48 | 64.37 | 61.46 | 66.64 |
| MassiveScenarioClassification | 68.75 | 69.05 | 66.41 | 70.38 |
| ToxicConversationsClassification | 59.14 | 67.28 | 66.90 | 62.85 |

Table 22: MTEB English classification results.

| Dataset | DAP | SONAR | LaBSE | MEXMA |
|---|---|---|---|---|
| Average | 67.46 | 63.13 | 68.69 | 66.25 |
| AmazonReviewsClassification (zh) | 34.35 | 31.91 | 32.98 | 33.40 |
| MassiveIntentClassification (zh-CN) | 71.99 | 62.08 | 63.86 | 74.41 |
| MassiveScenarioClassification (zh-CN) | 65.45 | 68.88 | 70.85 | 65.28 |
| JDReview | 71.54 | 69.59 | 79.13 | 70.73 |
| MultilingualSentiment | 62.03 | 57.69 | 65.52 | 60.34 |
| OnlineShopping | 85.03 | 75.64 | 85.62 | 80.09 |
| Waimai | 81.82 | 76.12 | 82.85 | 79.52 |

Table 23: MTEB Chinese classification results.

| Dataset | DAP | SONAR | LaBSE | MEXMA |
|---|---|---|---|---|
| Average | 63.76 | 61.88 | 62.05 | 66.07 |
| AmazonReviewsClassification | 35.60 | 34.91 | 38.52 | 35.62 |
| MTOPDomainClassification | 84.43 | 86.19 | 84.14 | 86.70 |
| MTOPIntentClassification | 65.78 | 66.75 | 62.01 | 74.12 |
| MassiveIntentClassification | 64.51 | 58.55 | 60.47 | 65.59 |
| MassiveScenarioClassification | 68.50 | 63.02 | 65.1 | 68.31 |

Table 24: MTEB French classification results.

| Dataset | DAP | SONAR | LaBSE | MEXMA |
|---|---|---|---|---|
| Average | 52.27 | 54.01 | 49.53 | 55.38 |
| DanishPoliticalCommentsClassification | 36.44 | 37.59 | 38.69 | 38.75 |
| LccSentimentClassification | 58.27 | 54.27 | 50.07 | 52.40 |
| MassiveIntentClassification (da) | 58.74 | 62.03 | 58.25 | 65.75 |
| MassiveScenarioClassification (da) | 66.15 | 67.76 | 65.24 | 69.26 |
| NordicLangClassification | 41.73 | 48.40 | 35.38 | 50.74 |

Table 25: MTEB Danish classification results.

| Dataset | DAP | SONAR | LaBSE | MEXMA |
|---|---|---|---|---|
| Average | 51.58 | 55.59 | 50.76 | 58.08 |
| MassiveIntentClassification | 55.85 | 59.90 | 57.91 | 64.48 |
| MassiveScenarioClassification | 62.67 | 65.81 | 64.29 | 68.22 |
| NoRecClassification | 46.06 | 48.25 | 45.44 | 48.88 |
| NordicLangClassification | 41.73 | 48.40 | 35.38 | 50.74 |

Table 26: MTEB Norwegian classification results.

| Dataset | DAP | SONAR | LaBSE | MEXMA |
|---|---|---|---|---|
| Average | 53.03 | 55.09 | 56.00 | 57.09 |
| AllegroReviews | 31.58 | 29.62 | 34.89 | 31.09 |
| MassiveIntentClassification (pl) | 58.53 | 65.86 | 59.71 | 66.85 |
| MassiveScenarioClassification (pl) | 63.05 | 69.99 | 64.58 | 70.20 |
| PAC | 67.97 | 73.87 | 68.11 | 73.31 |
| PolEmo2.0-IN | 61.75 | 52.80 | 64.00 | 59.10 |
| PolEmo2.0-OUT | 35.32 | 38.40 | 44.72 | 42.00 |

Table 27: MTEB Polish classification results.

| Dataset | DAP | SONAR | LaBSE | MEXMA |
|---|---|---|---|---|
| Average | 63.87 | 70.73 | 69.75 | 74.39 |
| PawsX | 55.30 | 75.05 | 54.07 | 73.18 |
| SprintDuplicateQuestions | 72.47 | 77.08 | 89.26 | 86.89 |
| XNLI | 63.83 | 60.06 | 65.92 | 63.10 |

Table 28: MTEB English pair classification results.

| Dataset | DAP | SONAR | LaBSE | MEXMA |
|---|---|---|---|---|
| Average | 73.03 | 77.57 | 73.70 | 78.13 |
| PawsX (fr) | 55.57 | 71.36 | 54.63 | 71.07 |
| Opusparcus (fr) | 100.00 | 100.00 | 100.00 | 100.00 |
| XNLI | 63.52 | 61.34 | 66.48 | 63.32 |

Table 29: MTEB French pair classification results.

| Dataset | DAP | SONAR | LaBSE | MEXMA |
|---|---|---|---|---|
| Average | 61.12 | 60.80 | 61.95 | 62.12 |
| PawsX(zh) | 56.20 | 65.35 | 54.26 | 63.68 |
| Cmnli | 69.29 | 61.86 | 72.67 | 67.45 |
| Ocnli | 57.86 | 55.18 | 58.91 | 55.23 |

Table 30: MTEB Chinese pair classification results.

| Dataset | DAP | SONAR | LaBSE | MEXMA |
|---|---|---|---|---|
| Average | 67.45 | 67.24 | 70.93 | 70.62 |
| BIOSSES | 70.51 | 79.11 | 78.70 | 75.97 |
| SICK-R | 69.18 | 62.94 | 69.99 | 66.00 |
| STS12 | 64.69 | 65.46 | 65.08 | 67.32 |
| STS13 | 63.50 | 62.79 | 67.98 | 67.05 |
| STS14 | 61.49 | 57.54 | 64.03 | 62.73 |
| STS15 | 75.38 | 74.25 | 76.59 | 75.72 |
| STS16 | 68.00 | 75.73 | 72.98 | 76.93 |
| STS17 (en-en) | 77.03 | 79.94 | 79.45 | 80.97 |
| STS22 (en) | 53.38 | 47.12 | 60.97 | 57.11 |
| STSBenchmark | 69.39 | 67.39 | 72.25 | 73.53 |
| STSBenchmarkMultilingualSTS (en) | 69.39 | 67.39 | 72.25 | 73.53 |

Table 31: MTEB English STS results.

| Dataset | DAP | SONAR | LaBSE | MEXMA |
|---|---|---|---|---|
| Average | 45.31 | 42.15 | 47.50 | 51.56 |
| ATEC | 28.01 | 26.18 | 26.61 | 29.68 |
| BQ | 40.01 | 37.66 | 42.60 | 44.37 |
| LCQMC | 54.97 | 50.11 | 52.19 | 61.34 |
| PAWSX | 12.99 | 32.75 | 10.23 | 27.77 |
| STS22(zh) | 52.05 | 52.82 | 63.02 | 63.49 |
| STSB | 63.67 | 50.18 | 68.38 | 65.75 |
| STSBenchmarkMultilingualSTS (zh) | 65.46 | 45.33 | 69.50 | 68.55 |

Table 32: MTEB Chinese STS results.

| Dataset | DAP | SONAR | LaBSE | MEXMA |
|---|---|---|---|---|
| Average | 67.74 | 65.60 | 74.33 | 70.10 |
| SICKFr | 66.84 | 66.1 | 69.94 | 65.94 |
| STS22 (fr) | 64.44 | 61.72 | 77.95 | 72.19 |
| STSBenchmarkMultilingualSTS (fr) | 71.92 | 68.99 | 75.1 | 72.17 |

Table 33: MTEB French STS results.

| Dataset | DAP | SONAR | LaBSE | MEXMA |
|---|---|---|---|---|
| Average | 57.06 | 57.17 | 65.82 | 63.67 |
| CDSC-R | 74.12 | 85.77 | 85.53 | 85.95 |
| SICK-R-PL | 60.63 | 62.98 | 65.90 | 64.31 |
| STS22 (pl) | 28.16 | 25.31 | 39.28 | 32.51 |
| STSBenchmarkMultilingualSTS (pl) | 65.31 | 54.62 | 72.58 | 71.93 |

Table 34: MTEB Polish STS results.

## F   TOKEN LEVEL ANALYSIS

In this section, we illustrate the behaviour of each model by visualizing the closest tokens in the space. We observe that MEXMA matches tokens in translations but also different contexts when tokens are used with the same meaning. This is further broken down in Table 9, which distinguishes between two types of matches MEXMA does: (1) "same language" matches, where the model identifies the same token used in a different context (monolingual), and (2) "other" matches, where it recognizes translated tokens in a sentence in another language that is not a translation (multilingual). We observe that SONAR primarily matches tokens across translations, but does not tend to match the same token when it appears in different sentences within the same language. Examples of MEXMA and SONAR comparisons of matching the same token in other sentences is in Figure 5, and both models matching translations in Figure 6. In both figures, we show the three closest tokens to the selected token, denoted as query on the green box, with the blue text. The closest tokens are in the purple boxes with the pink text. Additionally, we show examples of how LaBSE and MEXMA without direct token-level gradients (no-tok MEXMA), match adjacent tokens in the same sentence regularly. These are shown for LaBSE in Figure 2, and for no-tok MEXMA in Figure 3. Lastly, we show how XLM-RoBERTa mostly matches the same tokens in other sentences in the same language, presented in Figure 4. For these last three models, we show the top-2 closest tokens, with the same color scheme as mentioned above. Each image has two examples for the given model.

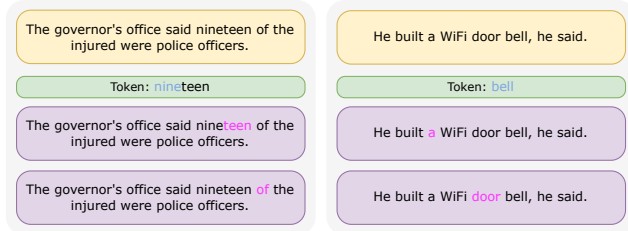

Figure 2: Example of LaBSE's token matching. The token in blue is the query token, the tokens in pink are the closest tokens to the query token in the space.



Figure 3: Example of MEXMA no token-level grad's token matching. The token in blue is the query token, the tokens in pink are the closest tokens to the query token in the space.

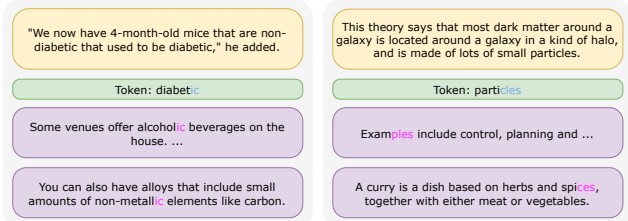

Figure 4: Example of XLM-RoBERTa token matching. The token in blue is the query token, the tokens in pink are the closest tokens to the query token in the space.

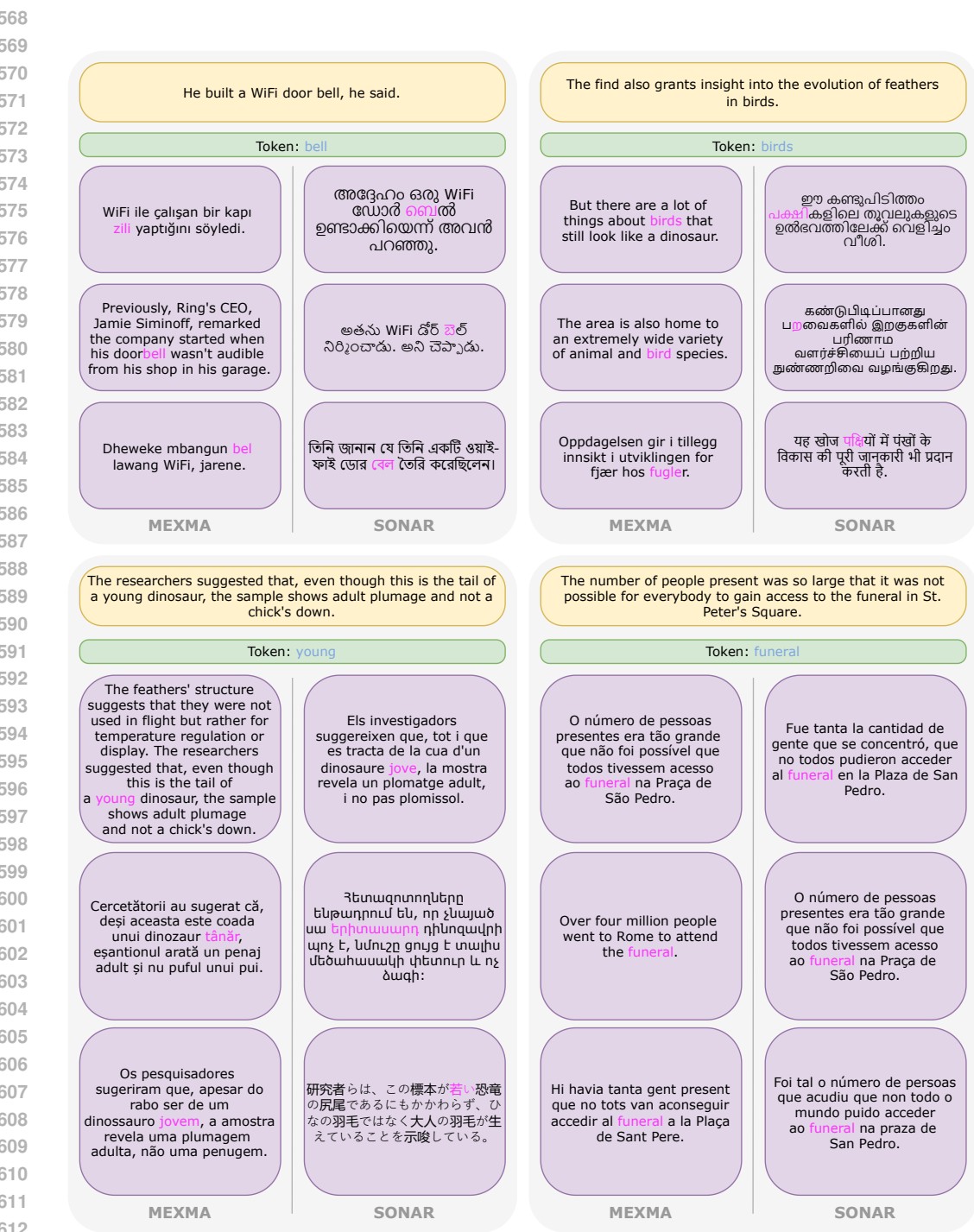

Figure 5: Comparison of SONAR and MEXMA token matching. MEXMA displays the ability to match a token in another sentence in the same language. SONAR matches a translated token. The token in blue is the query token, the tokens in pink are the closest tokens to the query token in the space. MEXMA is on the left, SONAR on the right.

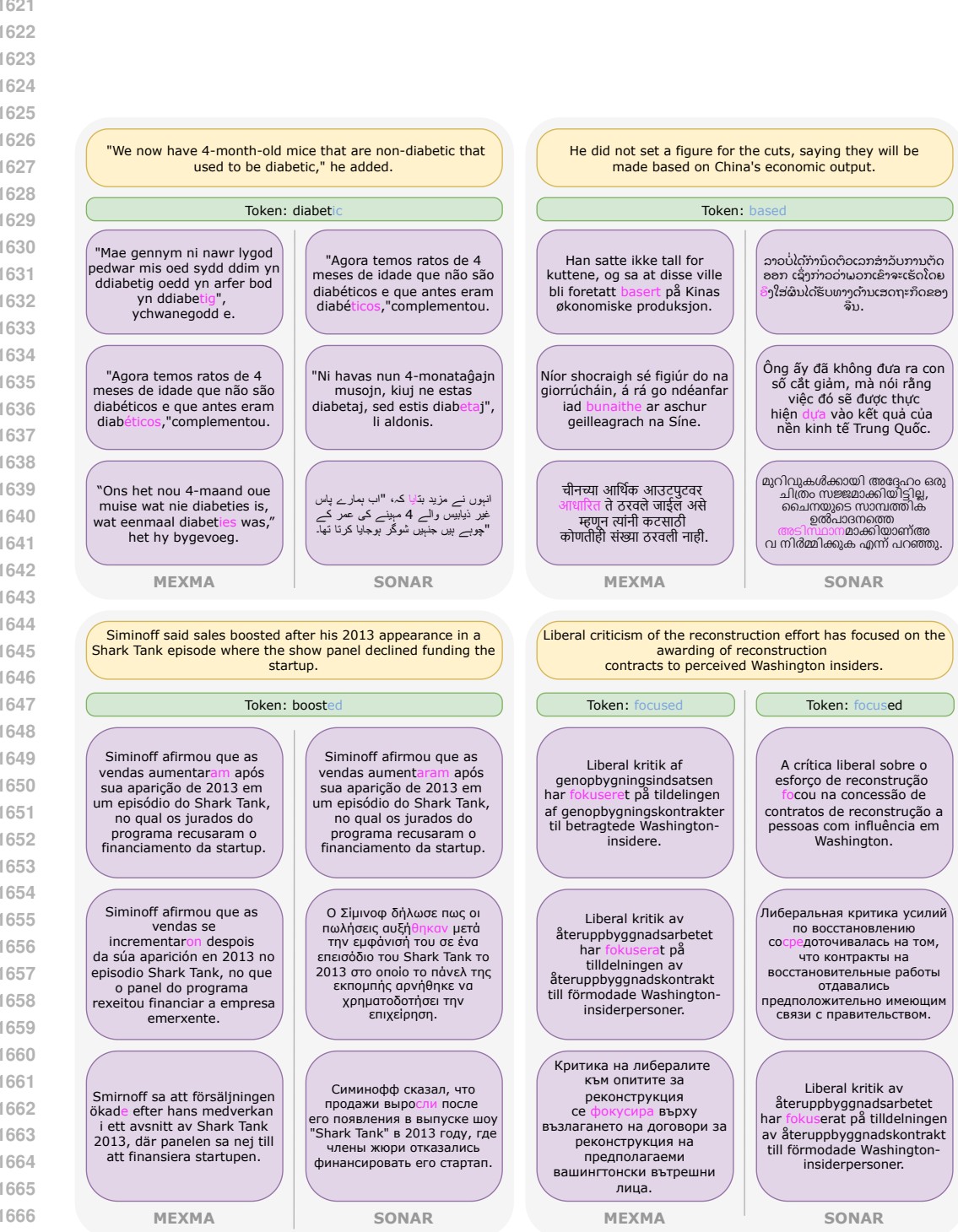

Figure 6: Comparison of SONAR and MEXMA on translated tokens in translations.

### F.1 VERIFYING THE TOKEN ALIGNMENT QUALITY THROUGH SIMALIGN

| Dataset | XLM-R | SONAR | LaBSE | MEXMA |
|---------|-------|-------|-------|-------|
| Average | 56.90 | 70.66 | 67.53 | 73.03 |
| eng-deu | 61.50 | 77.10 | 72.90 | 80.50 |
| eng-fra | 71.20 | 89.20 | 84.40 | 91.50 |
| eng-ces | 38.00 | 45.70 | 45.30 | 47.10 |

Table 35: SimAlign results using different models as backbone for the token-level alignment on different language pairs.

Several approaches have showed that aligned tokens across languages lead to better performing sentence representations (Li et al., 2023; Cao et al., 2020; Schuster et al., 2019). In order to further validate the improved alignment of our tokens, we use XLM-RoBERTa, SONAR, LaBSE and MEXMA as the backbone for SimAlign (Jalili Sabet et al., 2020). We test the models across 3 language pairs, on the datasets reference in SimAlign, English-German (Koehn, 2005), English-French Och & Ney (2000) and English-Czech Mareček (2008). The alignment created by the models is compared a reference word alignment to compute a F1 score. The results are provided in Table 35, and it is possible to see that the alignment created by MEXMA achieves better F1 scores. All results were achieved using the itermax method, taking the word representations from the last layer of each model.

## G  ATTENTION DISTRIBUTION OVER TOKENS

In this section, we provide some examples of MEXMA and LaBSE's attention probabilities given by the CLS token to the word tokens. The examples are provided in Figures 7, 8, 9 and 10. Across all figures, it is possible to see that LaBSE tends to be more uniform across all tokens, while MEXMA tends to focus more attention on a smaller subset of the tokens. All examples are taken from the FLORES200 test set with the xsim++ extension, where some words in the original sentences are replaced, and the models have to be able to still match the correct translation, and not a sentence with a small change. From Figure 7 to Figure 8 "nineteen" is replaced with "twenty nine". From Figure 9 to Figure 10 the word "white" is replaced with "black".

Figure 7 shows the attention placed by MEXMA and LaBSE on the same sentence in English and Portuguese. It is possible to see that MEXMA in Portuguese places most of the attention in two tokens, "governador" and "19", where the token in "19" is very relevant here since it is the one needed to distinguish the examples in xsim++. LaBSE seems to have many tokens with a lot of attention, and does not have "19" as one of the tokens with the most attention.

In Figure 8, we have the English example with nineteen (as previously shown in Figure 7) compared to the same sentence with nineteen replaced by twenty-nine. Interestingly, LaBSE places more attention on the "##teen" token than the "nine" token, but similar attention to the "twenty", "-" and "nine" tokens. MEXMA places similar attention in all nineteen tokens, and in twenty nine it places a small amount of attention on the irrelevant "-", with a higher degree of attention in "nine" and a smaller amount of attention in "twenty". MEXMA also seems to do a good job ignoring irrelevant tokens like "of", while LaBSE places a lot of attention in it.

Figure 9 has the same sentence in English and Portuguese, where, in xsim++ the models need to be able to match the color "white" instead of other colors. It is possible to see that, for LaBSE, white is not one of the most relevant tokens in English, but for MEXMA it is, along with "television". In Portuguese the behavior is similar, the token "bran" in "esbranquiçada" has a large degree of attention from MEXMA, while for LaBSE is it not a token with a lot of attention, and "çada" which is a token that does not convey the idea of white, is the one with the most attention out of the 4 tokens of the word, for LaBSE. In Portuguese it is also noticeable that MEXMA gives a small amount of attention to most of the less relevant tokens, while LaBSE seems to have a lot more tokens with a high degree of attention.

Figure 10 shows the same English sentence as Figure 9, with the word white replaced with the word black. Interestingly, MEXMA's attention remains the same with black and white, while for LaBSE the token "black" seems to get less attention than the token "white". The remaining tokens get similar attention in both models.

Additionally, Figure 11, provides a comparison for MEXMA and LaBSE with the probabilities of all heads, and all tokens, using BertViz (Vig, 2019). It is possible to see that MEXMA places a lot of attention on the EOS token, , which is used as an attention dump, i.e. an irrelevant token that receives a very large attention probability, a common phenomena in transformers, as explored in Xiao et al. (2024); Darcet et al. (2024); Sun et al. (2024). This happens frequently with MEXMA. It is, again, possible to see the difference in uniformity for MEXMA and LaBSE, with LaBSE having a more uniform attention in the figure. If we remove the BOS and EOS tokens from the entropy computation, we now get an entropy of $\approx 3.5$ and $\approx 3$ for LaBSE and MEXMA, respectively. MEXMA's entropy increases, while LaBSE stays mostly similar, which shows that MEXMA indeed frequently uses the EOS token as a dump. However, MEXMA still has a lower entropy and a more skewed distribution over its word tokens, with or without BOS and EOS, as shown by the lower entropy and the Figures 7-10.

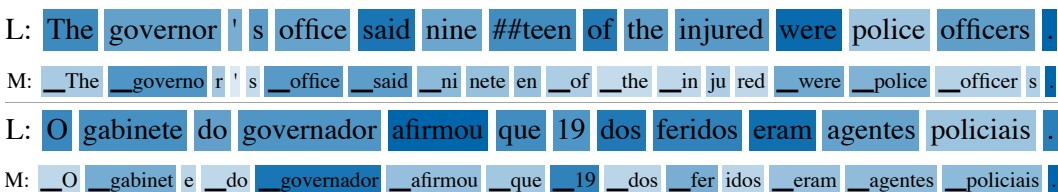

Figure 7: Comparison of LaBSE and MEXMA's probabilities distribution over the tokens. In this example, the models had to match the sentence with "19" in Portuguese and English. LaBSE's entries are preceeded with "L:", and MEXMA's with "M:".

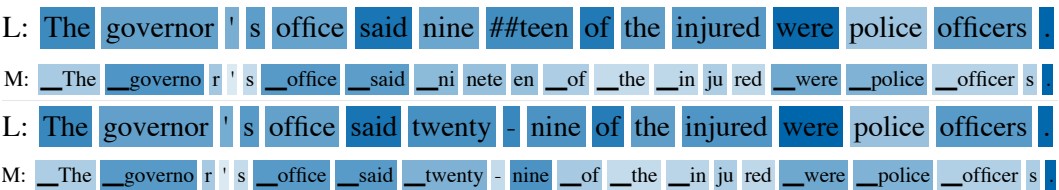

Figure 8: Comparison of LaBSE and MEXMA's probabilities distribution over the tokens. In this example, the models had to distinguish the sentence with "19" and "29" in Portuguese and English. LaBSE's entries are preceeded with "L:", and MEXMA's with "M:"

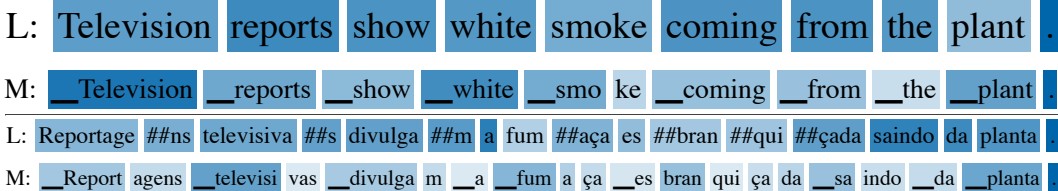

Figure 9: Comparison of LaBSE and MEXMA's probabilities distribution over the tokens. In this example, the models had to match the sentence with "white" in Portuguese and English. LaBSE's entries are preceeded with "L:", and MEXMA's with "M:"

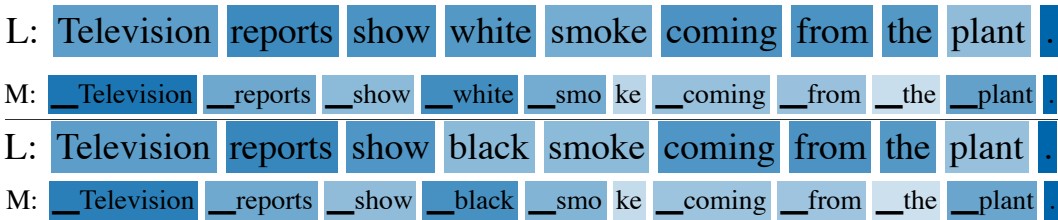

Figure 10: Comparison of LaBSE and MEXMA's probabilities distribution over the tokens. In this example, the models had to distinguish the sentence with "white" and "black" in Portuguese and English. LaBSE's entries are preceeded with "L:", and MEXMA's with "M:"

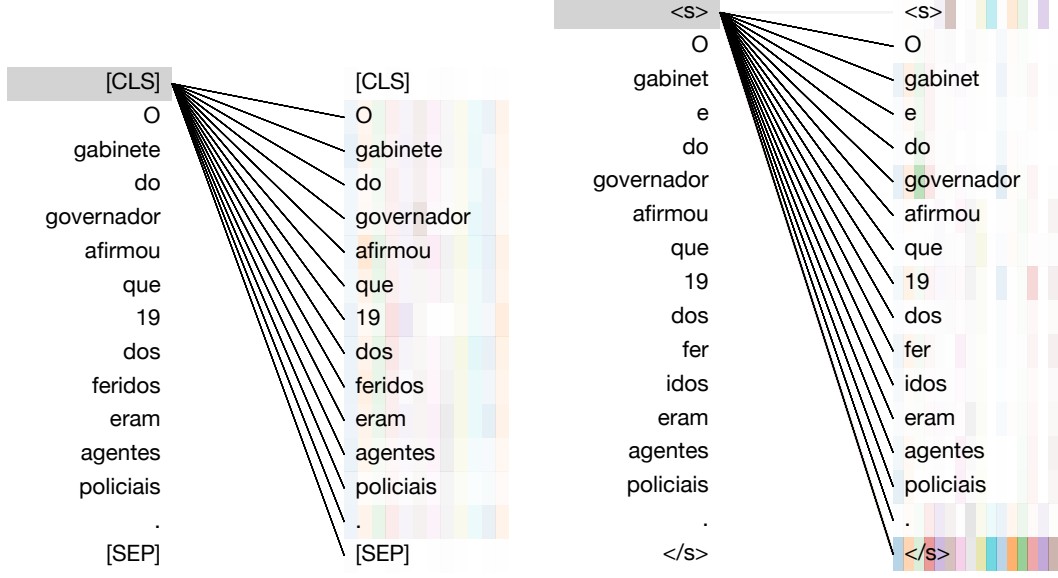

Figure 11: Attention distribution of MEXMA and LaBSE across all heads, and all tokens. On the left is LaBSE, on the right is MEXMA. MEXMA uses the EOS token as an attention dump, and has a more skewed distribution, while LaBSE has a more uniform distribution.

# H BASELINE ARCHITECTURES

We report SONAR, LaBSE's, DAP's and RetroMAE's architectures in Figures 12b, 12a, 12c and 12d, respectively for easier comparison. LaBSE employs a slightly modified contrastive loss, to increase separation, and SONAR is based on translation. DAP uses token-level objectives, but it does not leverage them to update the sentence representation. RetroMAE uses the sentence in the heavy unmasking, but that unmasking does not update the tokens outputted by the encoder, it is monolingual, and the sentence representation does not come from an unmasked input. MEXMA is based on cross unmasking and has direct token level gradients updating its internal representations.

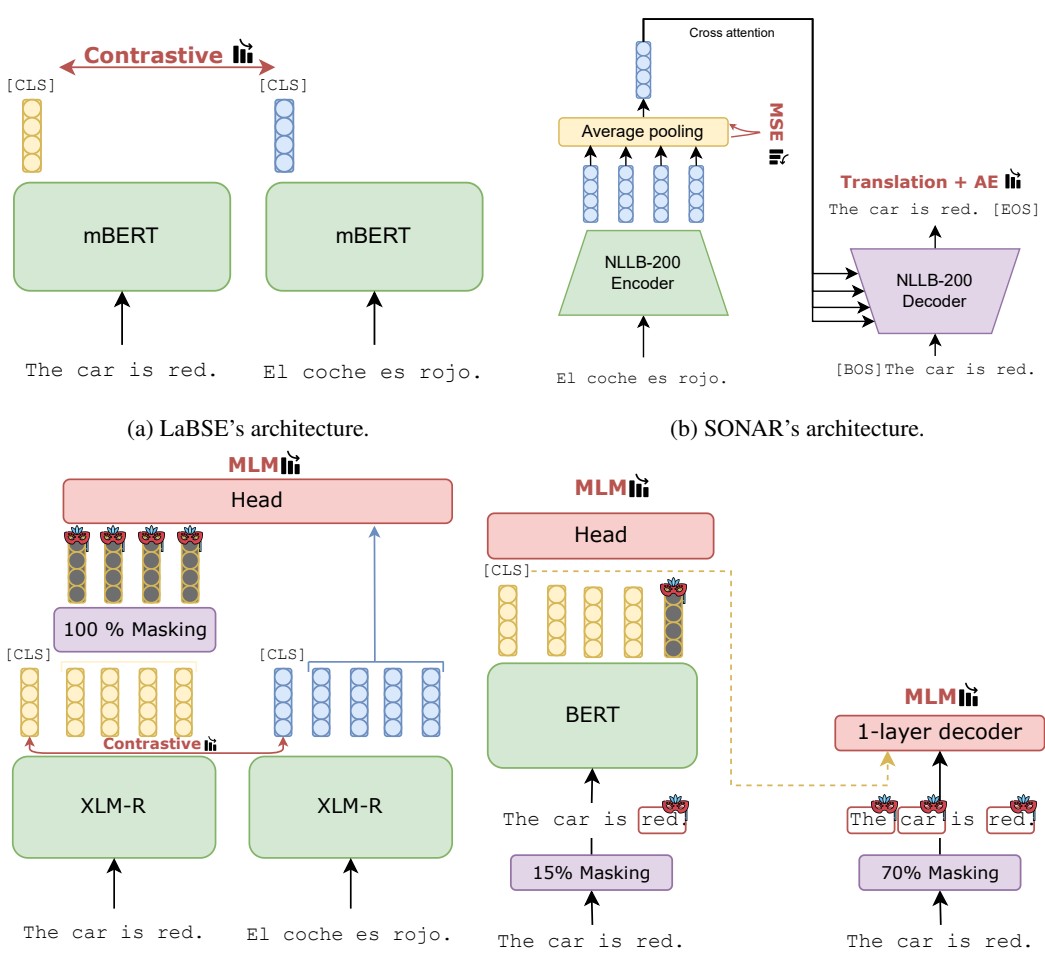

Figure 12: Architecture of the baselines.

