# OpenReview forum: "MEXMA: Token-level objectives improve sentence representations"
_ICLR.cc/2025/Conference — ICLR 2025 Conference Withdrawn Submission_

### Official Review · Reviewer_DqXv · 2024-10-25

**Soundness:** 2
**Presentation:** 2
**Contribution:** 2
**Rating:** 5
**Confidence:** 3

**Summary:**

This paper proposes a new cross-lingual sentence encoder (CLSE) called MEXMA. The contribution is the use of token-level objectives in addition to sentence-level objectives during training. Previous work such as LaBSE and SONAR use sentence-level objectives soley, e.g. making sure sentence vectors for translation pairs are close. The token-level objective is similar to a cross-entropy objective in masked language modeling, except that inputs for reconstruction come from the other language. The evaluation includes on sentence alignment tasks like xsim and downstream classification tasks like STS and MTEB.

**Strengths:**

1. Cross-lingual sentence encoders have many applications. It is worthy to explore ways to improve upon current methods.
2. The idea of incorporating cross entropy token loss in a cross-lingual way is interesting.

**Weaknesses:**

The main weakness is the results discussion: the majority of results in Section 4 and Section 5 focus on comparing the proposed MEXMA to previous approaches like LaBSE and SONAR. While I understand the goal to demonstrate competitiveness with "SOTA" methods, I found the discussion not too insightful. MEXMA, LaBSE, SONAR are all trained on different data using different pretrained models. There is no way to directly compare their task accuracies and make strong claims about the results, since there are too many underlying differences among systems. So I am not sure if I learned anything from the experiments and results discussion.

I have a few suggestions:

First, keep Table 1-4 which does help you establish SOTA results but vastly simplify the discussion. There is no need to go through detailed comparisons with the different methods, since as said before, they are not directly apples-to-apples comparisons.

Second, include a more comparable baseline in Tables 1-4: ideally, this should be the same backbone model and training data as your proposed MEXMA, but changing the objectives to be more similar to previous approaches. The contrastive MEXMA without token-level gradients in Table 6 is a good candidate. Or, you can re-implement the exact LaBSE or SONAR objective with your system. By presenting the baseline alongside the full results, it will be easier to see if there are improvements.

Third, the ablation analysis of Sec 5.1 and 5.2 is useful but can be expanded. I think it will be helpful to explain results in more detail. For example, what does it mean that "no token-level gradients": I understand technically what it means but I do not have a good intuition why that result is significant: the objective still has token-level CE loss but parts of the gradient have been stopped--what does that mean about the impact of the token-level loss?

**Questions:**

Questions:
1. Can you clarify whether the KoLEO loss is simply an add-on that is useful for all sorts of cross-lingual sentence encoders, or is anisotropy an particularly important problem in MEXMA?

2. Do you think the method would work with a CE loss that is monolingual (simply MLM objective on each language independently, perhaps similar to some previous work), rather than cross-lingual CE loss as proposed? Basically, I wonder whether the argument is that introducing any kind of token-level objective will help balance the sentence objective. Does it really need to be a cross-lingual objective?

Comment:
- L_{mlm} on page 4 might be hard to understand on face value. I understand what you meant but it might be confusing what kind of input is taken by CE([S_B,\hat{A}],A) -- outputs of encoders, or "concatenation" like [S_B,\hat{A}]? The CE() term is overloaded, I guess.

---

> ### Author Response · Authors · 2024-11-19
>
> We thank the reviewer for their comments and suggestions. It resulted in several additions to the paper, that we believe improved its quality.
>
> **Fair comparisons**:
>
> We appreciate the recommendation to enhance the fairness of the comparisons. Although we did not include it in the paper, as our primary focus was on presenting the optimal scores for each model, we did explore your suggestions and incorporated them into our analysis.
>
> For LaBSE, we did reimplement their training objective using XLM-R and our training data. However, since the xsim++ results were significantly worse than the ones we obtained with the public LaBSE model, we decided not to include them.You can find those results in the table below under “LaBSE with XLM-R”. Given there is no official public implementation of LaBSE, and the large gap in xsim++ compared to the public model, we opted to always compare it to the public model.
>
> For fairer comparison with SONAR, we trained MEXMA using the NLLB encoder as backbone. This is easier than training SONAR with XLM-R as backbone, since we would need a translation decoder for XLM-R, or initialize it randomly, which would be less fair to compare to SONAR that starts from NLLB, a very strong translation model. The results are worse than MEXMA and SONAR across all tasks. This result is likely due to the fact that NLLB was pre-trained with a translation objective, and was never trained to do masked language modeling, like XLM-R.
>
> We report here the results obtained with LaBSE/XLM-R and MEXMA/NLLB in the following table:
> | Model (always on our data) | xsim | xsim++ | SentEval |
> |----------------------------|------|--------|----------|
> | LaBSE with XLM-R           | 0.10 | 33.82  | 86.08    |
> | MEXMA with NLLB            | 0.11 | 23.36  | 85.20    |
> |                            |      |        |          |
> | Baselines                  |      |        |          |
> | MEXMA                      | 0.06 | 9.60   | 86.38    |
> | SONAR                      | 0.09 | 12.08  | 85.82    |
> | LaBSE                      | 0.92 | 18.65  | 85.63    |
>
> We thank the reviewer for these suggestions. These results have been added to the paper under appendix B.4.
>
> **Importance of token-level gradients**:
>
> The token-level gradient in MEXMA is both important and subtle. We try to clarify this concept in the following and will update the paper with a clearer explanation.
> Our experiments are designed to demonstrate the importance of how the encoder is updated, whether from token-level and/or sentence-level gradients, on overall performance. Specifically, when we apply only cross-entropy (CE) or combine CE with alignment while stopping the gradient flow at the token level, the encoder is updated solely through gradients derived from the sentence representation. Our goal is to illustrate that allowing gradients to flow from each masked token provides the encoder with a richer training signal, thereby enhancing its quality. The ablation study, which involves stopping the gradients, is intended to highlight this effect.
> We wish to underscore not merely the presence of a token-level loss, but the distinction between updating the encoder exclusively via sentence-level gradients versus incorporating gradients from all tokens. Our findings indicate that the inclusion of these additional gradients significantly boosts performance.
>
> **Does KoLeo loss apply to all CLSE**?
>
> The KoLeo loss is particularly relevant in MEXMA because we have a non-contrastive alignment loss, which tends to create a space that is more collapsed than a contrastive objective, similarly to DINOv2 (cite). We can say this according to a basic measure of the variance across components. Approaches like SONAR and MEXMA would benefit from the KoLeo loss. However, a contrastive approach like LaBSE would not, given that it already has "repulsing forces" that increase the spread over the space. Therefore, anisotropy is a more significant issue in MEXMA (and other non-contrastive approaches), making the KoLeo loss a valuable addition. Thank you for pointing out this was not clear enough, we will improve its clarity in the paper.
>
> **Monolingual cross-entropy, instead of cross-lingual**
>
> Based on our experiments the cross-lingual objective is crucial for the performance of the model. Without it the tokens are not as aligned across languages, and the sentence representations do not encode as much information. The end result is that this worsens performance. However, thank you for your suggestion, we have added a comparison of monolingual and cross-lingual unmasking in appendix B.2 of the paper, where we show that monolingual performs much worse.
>
>
> We thank the reviewer for their comments and suggestions, we believe they increased the quality of our paper. We added fairer comparisons with current SOTA in appendix B.4, and added an ablation of the importance of the cross-linguality in appendix B.2.

---

### Official Review · Reviewer_bgeV · 2024-10-30

**Soundness:** 4
**Presentation:** 4
**Contribution:** 3
**Rating:** 8
**Confidence:** 3

**Summary:**

This paper introduces MEXMA, a new symmetrical model architecture mixing sentence-level and token-level objectives for training multilingual sentence representations. More specifically, MEXMA is trained with a combination of three losses: 1) a token-level cross-unmasking loss in which the sentence representation of a sentence in one language is used to perform masked language modeling in its translated version; 2) An alignment loss between sentence representations across languages, following [SONAR](https://arxiv.org/abs/2308.11466); and 3) a [KoLeo loss](https://openreview.net/forum?id=SkGuG2R5tm) to promote representation isotropy. Authors conduct a detailed evaluation of their proposed approach, showing improved results in multilingual sentence alignment, single and pairwise sentence classification, with competitive results for semantic text similarity. Finally, several ablations are conducted to estimate the importance of individual model components, model size and masking ratios. Two analysis are also conducted to compare token representations properties across various models and the entropy of attention probabilities in MEXMA and SONAR, highlighting differences produced by the additional token-level MEXMA loss.

**Strengths:**

This work provides a valuable contribution by proposing a novel, intuitive, and effective approach to train sentence representations from multilingual aligned data. The paper's contents are structured well, and the method is presented clearly and concisely while also motivating its differences from previous approaches. The experiments are very comprehensive and show clear improvements across a diverse set of benchmarks. Ablations are performed thoroughly and further support the authors' design choices, confirming the usefulness of the proposed token-level loss for improving the quality of resulting sentence embeddings. The final analyses of Sections 5.5 and 5.6 provide an interesting overview of how the proposed training procedure practically affects the properties of learned representations. As a side note, the appendix provides extensive and useful additional materials, and the inclusion of detailed visualizations of baseline architectures was very helpful in illustrating the novelty of the proposed approach.

**Weaknesses:**

As a general comment, Section 5.6 was interesting, but I find it not very well motivated. It is unclear what useful information is to be gained by knowing about the lowest entropy of attention probabilities in MEXMA as opposed to LABSE. On the other hand, the attention sink phenomenon shown in Figure 11 seemed more interesting for readers interested in understanding mechanisms in the MEXMA model. Still, it was placed in the Appendix and never mentioned in the main body of the paper.

**Questions:**

Minor comments:

Line 457: It would be good to specify which size of the NLLB model is used in this experiment (several are available) and whether the encoder or decoder output representations are used in the comparison.
Line 525: Typo, loose should be lose

---

> ### Author Response · Authors · 2024-11-19
>
> Thank you for going through our paper thoroughly and in detail. We are glad that you appreciated our work and found it valuable.
>
> **Section 5.6 is not clear what we gain by knowing about the lowest entropy in MEXMA. The attention sink is more interesting for the paper and was placed in the appendix.**
>
> We are grateful for your recognition of the importance of Section 5.6. Our primary objective was to delve into the behavior of the models to elucidate why they yield varying results in downstream tasks. By examining both the content of the tokens (Section 5.5) and the manner in which these tokens are integrated to form sentence representations (Section 5.6), we gain a comprehensive understanding. Section 5.6, in particular, reveals that MEXMA's focuses on specific tokens, rather than attending uniformly to all tokens, underscoring its ability to highlight the most critical tokens for sentence representation.
> While the attention sink phenomenon is indeed intriguing, to preserve the focus of the paper, we decided not to involve too much of the main manuscript on it. However, as per your valid suggestion, we have referenced various relevant papers for readers that might be interested.
>
> **L457 we should specify the size of NLLB model**
>
> We apologize for the oversight and thank the reviewer for pointing this out. We had mentioned the size of the SONAR encoder in Table 7 (766M parameters), which led us to believe that specifying the size of NLLB was unnecessary (1.3B parameters). We appreciate your attention to detail and have added this information in the paper.
> Regarding the representations used for the comparison, we only use the output of the encoder to create the sentence representations, we do not use the decoder. We have also added this information in the paper to improve clarity.
>
> Thank you for your comments and suggestions for improvement, we have incorporated them into our paper.

---

### Official Review · Reviewer_CKqK · 2024-11-03

**Soundness:** 4
**Presentation:** 3
**Contribution:** 2
**Rating:** 5
**Confidence:** 4

**Summary:**

This paper proposes MEXMA, a framework for training sentence encoding models. MEXMA aims to overcome the limitations of existing cross-lingual sentence encoders that rely solely on sentence-level objectives. By integrating token-level objectives, MEXMA leverages the information encoded in individual tokens to enhance the quality of the sentence representation. The authors demonstrate that this approach significantly improves performance on bitext mining and some other downstream tasks, outperforming state-of-the-art models like SONAR and LaBSE.

**Strengths:**

- This is a well-written paper that introduces an effective framework for training cross-lingual sentence encoders.
- The performance is very strong, and I foresee that MEXMA will be widely adopted in various tasks that need cross-lingual sentence representations.

**Weaknesses:**

- My major concern is the novelty. In fact, the lexical-level approach is not as novel as the authors claim. This ICLR 2020 paper (https://openreview.net/forum?id=r1xCMyBtPS) and this NAACL 2019 paper (https://aclanthology.org/N19-1162/), neither mentioned in the submission, have explored very similar post-training fine-tuning ideas to improve cross-lingual sentence/contextualized word representations.
All the significant components are adapted from existing work.
Considering the above, combined with the significant improvement in results, a better fit for this work would be the industrial/demo track of a conference rather than a main conference that aims at more scientific contributions.

- Massive high-quality supervised bitext data is required for training MEXMA.
This restricts possible model generalization, as such data is not always available for low-resource languages.
This also limits the method generalization to multimodal settings, as in such settings, modalities are more naturally complementary instead of being parallel.

- The analysis of the paper could be improved. See my questions for more details.

**Questions:**

- In L171 equation, it seems a natural alternative is concatenating the detailed token-level embeddings of A and B in both parts of the objective. Do you have specific intuition on why only concatenating the sentence-level embeddings to the counterpart?
- I understand there's limited space to show the language-specific results of Table 1, but it would be very helpful to see the results in the supplementary material.
- Another interesting analysis would be comparing your model and XLM-R as backbones of SimAlign (https://aclanthology.org/2020.findings-emnlp.147/), which considers automatic word-level alignment extraction from pre-trained cross-lingual sentence encoders. The results will provide insights into the effectiveness breakdown of MEXMA.

---

> ### Author Response · Authors · 2024-11-19
>
> Thank you for the thorough comments and suggestions, we believe that they resulted in the addition of relevant experiments, related work and strengthened our paper.
>
> **Novelty**:
>
> We appreciate the mention of those relevant papers.
> The first mentioned paper https://openreview.net/forum?id=r1xCMyBtPS shows that aligned word representations result in better sentence performance across languages, which is very relevant for our work indeed, as shown by DAP that we cite in our paper. We now cite those two papers, thanks for the suggestion.
> The objective of the mentioned papers is, however, different from ours, since they aim at aligning word representations, while our objective is to create general sentence representations that encode the relevant information of the sentences and are aligned across languages.
>
> However, compared to existing Cross-lingual sentence encoder (CLSE) approaches, combining both MLM and alignment or not, our approach differs from previous existing approaches in the following key novel aspects:
> * Previous approaches did not use unmasking as a means to create a better sentence representation as we do, with heavy token unmasking given the sentence representation, enforcing the sentence to encompass all the relevant information in the input. Previous approaches also don’t mix tokens in one language and sentence representations in another. We show this is crucial for the performance of MEXMA, and added an ablation in appendix B.2 that shows the importance of cross-linguality for performance.
> * Another relevant point is that most previous approaches updated the encoder only with a gradient from the sentence representation, meanwhile we update the encoder with gradients from both the sentence and the individual tokens. We show in sections 5.1 and 5.2 that this change is crucial to achieve our new SOTA results, increasing performance across all tasks. Specifically, it improved the xsim++ results in 1.7% (taken from table 5 in the paper).
> * Additionally, most approaches are contrastive which require large batch sizes and/or good hard negatives to work well. In MEXMA (and SONAR), this is not needed because of the usage of a non-contrastive alignment loss.
>
> **Massive quality supervised data**:
>
> We acknowledge the reviewer's concern regarding the dependency on large datasets in SOTA approaches. It is indeed a common characteristic among current SOTA methods, such as SONAR, LaBSE, and DAP, to require substantial amounts of translation data. Many of these approaches, namely LaBSE, utilize datasets that are either difficult to obtain or process. In contrast, our approach leverages publicly available data, specifically a subset of the data used to train SONAR.
> The dataset we employ is described in the NLLB paper [1], which released high-quality data for 200 languages, including many low-resource languages. The availability and quality of the NLLB data facilitate the extension of our work to additional low-resource languages or the augmentation of existing language datasets with relative ease.
> We believe that the reliance on large datasets will continue to be a significant factor for SOTA approaches in the foreseeable future. Our approach, by utilizing accessible and high-quality data, aims to address this challenge and contribute to the ongoing development of multilingual models. This strategy not only supports the scalability of our method but also underscores its potential applicability to a broader range of languages.
>
>
> **Multimodality**:
>
> Extending the work to multiple modalities is out of the scope of this paper, given that our objective was to create multilingual sentence representations from text. If we were to include multiple modalities we would have to consider different encoders, losses and strategies, namely CLIP-like. However the MEXMA architecture would be suitable to replace a CLIP-like approach, i.e. an approach where a contrastive loss is applied between a text and image encoder.
>
>
> **Analysis**:
> * __Concatenating tokens instead of using the sentence representation as context__
>      1. If we concatenate the tokens of A and B, in a similar manner to XLM-R, e.g. [CLS] t_a_1, t_a_2, … , t_a_n [SEP] t_b_1, t_b_2, … , t_b_n [SEP], we risk not ensuring that the sentence representation captures all relevant information from the input. In this setup, the masked tokens would be contextualized by all other tokens in another language, rather than focusing on the sentence representation, as is done in MEXMA. While DAP employs a method akin to your suggestion and performs quite effectively, our results across several tasks indicate that our approach more effectively enforces the sentence representation to encode the necessary input information.
> * __Language specific results__
>    1. We add the results in appendix E.1.

---

> > ### Author Response · Authors · 2024-11-19
> >
> > Continuation:
> >
> > **Adding SimAlign analysis**
> >
> > We ran SimAlign with XLM-R, MEXMA, LaBSE and SONAR as backbones for half on the datasets mentioned in their repository (english-german, english-french and english-czech). We use the word representations from the last layer as in our analysis, and the itermax method from SimAlign:
> > | Dataset        | XLM-R | SONAR | LaBSE | MEXMA |
> > |----------------|-------|-------|-------|-------|
> > | Average        | 56.90 | 70.66 | 67.53 | 73.03 |
> > | english-german | 61.50 | 77.10 | 72.90 | 80.50 |
> > | english-french | 71.20 | 89.20 | 84.40 | 91.50 |
> > | english-czech  | 38.00 | 45.70 | 45.30 | 47.10 |
> >
> > The results seem to further indicate the good token alignment created by MEXMA. We thank the reviewer for the suggestion, we have added these results in the paper in appendix F.1, which makes our paper stronger by making the improvements in token alignment clearer with a more straightforward metric. Since SimAlign uses reference word alignments to compare to the token alignments created by the different models, it allows us to easily compute a single accuracy metric.
> >
> >
> >
> > We thank the reviewer again for his comments, we believe we were able to strengthen our paper with the suggestions. We added the full results for all languages for the mining tasks under appendix E.1, added additional alignment metrics based on SimAlign in appendix F.1, as well as the proposed references, and we added an analysis to ablate the importance of the cross-lingual unmasking under appendix B.2.
> >
> > [1] Costa-jussà, Marta R., et al. "No language left behind: Scaling human-centered machine translation." arXiv preprint arXiv:2207.04672 (2022).

---

### Official Review · Reviewer_xwWb · 2024-11-04

**Soundness:** 2
**Presentation:** 3
**Contribution:** 2
**Rating:** 3
**Confidence:** 5

**Summary:**

The paper presents MEXMA, a novel approach to enhancing cross-lingual sentence encoders (CLSE) by integrating both sentence-level and token-level objectives. Traditional CLSE methods focus solely on sentence-level objectives, which can result in information loss, particularly at the token level, and negatively impact sentence representation. MEXMA addresses this issue by using sentence representations in one language to predict masked tokens in another language, with updates directly influencing both the sentence representation and all tokens within the encoder. This incorporation of token-level objectives significantly enhances the quality of sentence representations, leading to improved performance in bitext mining and various downstream tasks compared to existing pre-trained CLSE models. The study also explores the information encoded in tokens and how they contribute to the construction of sentence representations.

**Strengths:**

1. This paper proposes a multi-grained training objective for fine-tuning a pre-trained language model to enhance cross-lingual sentence representation. The method is straightforward to reproduce.

2. Experimental results demonstrate that incorporating token-level objectives into the training of cross-lingual sentence encoders (CLSE) significantly improves the quality of sentence representations, surpassing the performance of current state-of-the-art pre-trained CLSE models in bitext mining and other downstream tasks.

**Weaknesses:**

1. The concept of learning both sentence-level and token-level alignment has been explored in several studies, such as those by Wei et al. (2021) and Fan et al. (2022). The authors should clarify the differences and advantages of their proposed method compared to these previous works.

2. There are several cross-lingual benchmarks, such as XTREME, that could be used for comparison. The authors are encouraged to evaluate their method against other works using these benchmarks and provide a detailed analysis of their proposed approach in relation to other methods concerning cross-lingual representation.




Wei et al., 2021. On learning universal representations across languages

Fan et al., 2022. Sentiment-aware word and sentence level pre-training for sentiment analysis

**Questions:**

N/A

---

> ### Author Response · Authors · 2024-11-19
>
> Thank you for your comments on the paper, and for pointing out  some relevant papers.  We added them in the related work section.
>
> The first paper “Wei et al., 2021. On learning universal representations across languages”, as Mexma, combines masked language modeling with an alignment loss. However there are several key differences:
> * (Wei et al., 2021) uses a contrastive loss to align sentences. Although it is the most common alignment loss in Cross Lingual Sentence Encoders (CLSE), using it effectively requires large batch sizes or careful hard negatives [1,3,7]. The usage of non-contrastive losses makes training easier [3,4,5,8], since they do not have those restrictions, however they are prone to collapse (producing the same output regardless of the input). Using a Masked Language Modeling (MLM) loss on the tokens prevents this collapse, so we use a non-contrastive loss, making it less reliant on large batch sizes or hard negatives and easier to train. Although (Wei et al., 2021) also has a MLM loss, they still rely on the contrastive loss to create the sentence alignment.
> * In (Wei et al., 2021) the sentence representations that are used for alignment are based on masked inputs, which creates a mismatch between training and inference, since at inference time tokens are not masked. This mismatch degrades inference time performance, as shown in ELECTRA [6]. In MEXMA the alignment between sentences is done between two sentence representations coming from an unmasked input, which eliminates this mismatch. Table 11 in our paper shows the improvements we get from using the symmetrical architecture without mismatch, compared to an alignment with masked representations. We get an improvement of 0.72% xsim++ going from aligning a masked to a non-masked sentence representation (which should already be better than aligning two masked representations) to a non-masked to non-masked alignment.
> * The unmasking task in (Wei et al., 2021) is used to keep the token quality similar to the one in the backbone pre-training. However, in MEXMA the unmasking task is used as a way to improve the quality of the sentence representation, by using the sentence in a different language as the context for token unmasking. The improvement brought by  this change is now presented  in appendix B.2 for completeness.
> * (Wei et al., 2021) uses a contrastive loss to approximate the sentence representation to the token representations (and vice versa). Although this does promote token alignment across languages, it also imposes a hard restriction on the sentence representation. In MEXMA, we instead use the unmasking head based on the sentence representation in another language to align the tokens across languages by transitivity, without introducing hard constraints in the sentence representation. This allows the sentence representation to take the best values necessary for the alignment and MLM losses.
>
> More generally, the key difference is that the concept in MEXMA is using the sentence representation in one language to allow the unmasking of tokens in another language, which forces the sentence representation to encode all of the information in the input, and enable representations to be more language independent, and aligned. This concept is not present in either of the mentioned approaches, and is key to our approach.
>
> Furthermore, a direct comparison is challenging due to the absence of a public model checkpoint or open-source code. However, we can still provide a comparison based on the results reported for the Tatoeba mining task in XTREME, alongside the results from LaBSE. LaBSE reports an accuracy of 83.7%, while (Wei et al., 2021) reports 69.1%. Considering that our approach has consistently outperformed LaBSE across all tested mining tasks, and noting the significant difference between the results of (Wei et al. , 2021) and LaBSE, it is reasonable to anticipate that our method also surpasses (Wei et al., 2021). To facilitate replication and future comparisons, we will open source our training code and release a model checkpoint.
>
> The second mentioned paper (Fan et al., 2022) is also very interesting and related to our work since it combines the usage of masked language modeling with an alignment loss. We thank this suggestion and have added it in our related work. Some key differences:
> * The approach uses information specific to the sentiment analysis task, which is very interesting, although it is orthogonal to our approach, since our goal is to create task agnostic representations, and keep our method generic.
> * Their model creates representations that excel at the task of sentiment analysis, but introduces constraints that might worsen the performance in other unrelated tasks. We show in our paper that MEXMA performs well on a wide array of tasks.

---

> > ### Author Response · Authors · 2024-11-19
> >
> > Continuation:
> >
> > Regarding evaluation, we believe our existing evaluations on the newer MTEB benchmark already cover most of the task types in XTREME: single sentence classification, sentence pair classification and bitext mining. MTEB comprises more datasets in each task, making the evaluation more robust. For instance, XTREME only uses NLI and PawsX for their sentence pair classification, meanwhile in MTEB it has PawsX, SprintDuplicateQuestions, XNLI, Opusparcus, Cmnli, Ocnli. We agree with the reviewer that using cross lingual benchmarks is of extreme importance to assess the quality of such models, and thank you for highlighting the importance of using such benchmarks.
> >
> > We thank the reviewer again for the comments that we believe allowed us to strengthen our work, with new references and comparisons in the related work, and a new analysis that allowed us to better emphasize one of the key points in our model, added under appendix B.2. We would also like to ask if the reviewer has any further questions, remarks or suggestions on how we can improve the paper further.
> >
> >
> > [1] Wei, Xiangpeng, et al. "On learning universal representations across languages." arXiv preprint arXiv:2007.15960 (2020).
> > [2] Fan, Shuai, et al. "Sentiment-aware word and sentence level pre-training for sentiment analysis." arXiv preprint arXiv:2210.09803 (2022).
> > [3] Bardes, Adrien, Jean Ponce, and Yann LeCun. "Vicreg: Variance-invariance-covariance regularization for self-supervised learning." arXiv preprint arXiv:2105.04906 (2021).
> > [4] Oquab, Maxime, et al. "Dinov2: Learning robust visual features without supervision." arXiv preprint arXiv:2304.07193 (2023).
> > [5] Duquenne, Paul-Ambroise, Holger Schwenk, and Benoît Sagot. "SONAR: sentence-level multimodal and language-agnostic representations." arXiv e-prints (2023): arXiv-2308.
> > [6] Clark, K. "Electra: Pre-training text encoders as discriminators rather than generators." arXiv preprint arXiv:2003.10555 (2020).
> > [7] Radford, Alec, et al. "Learning transferable visual models from natural language supervision." International conference on machine learning. PMLR, 2021.
> > [8] Chen, Xinlei, and Kaiming He. "Exploring simple siamese representation learning." Proceedings of the IEEE/CVF conference on computer vision and pattern recognition. 2021.

---

### Author Response · Authors · 2024-11-27

We appreciate the reviewers' time and effort in providing feedback on our submission. We have responded to all comments and uploaded a revised manuscript (with changes in blue). We are following up to ensure your concerns have been addressed. Any further questions or concerns - we would happily address them.

---

### Note · Authors · 2025-02-12

I have read and agree with the venue's withdrawal policy on behalf of myself and my co-authors.

---

### Meta-Review · Area_Chair_B49h · 2024-12-20

**Metareview:**

The paper proposes a cross-lingual sentence representation method by combining both sentence-level and token level objectives.

The paper receives inconsistent scores from reviewers. I concur with two of the reviewers who raised concerns about the novelty and comparison against previous work. The authors provided a response pointing out certain difference between the paper and previous work but I do not feel it fully addressed the concern.

**Additional Comments On Reviewer Discussion:**

The paper receives inconsistent scores from reviewers. I concur with two of the reviewers who raised concerns about the novelty and comparison against previous work. The authors provided a response pointing out certain difference between the paper and previous work but I do not feel it fully addressed the concern.

---

### Decision · Program_Chairs · 2025-01-22

Reject